# Wildlife Emergency Response Services Data Provide Insights into Human and Non-Human Threats to Wildlife and the Response to Those Threats

Elodie C. M. Camprasse [1,*], Matthias Klapperstueck [2] and Adam P. A. Cardilini [1]

1   School of Life and Environmental Sciences, Melbourne Burwood Campus, Deakin University, 221 Burwood Highway, Burwood, VIC 3125, Australia
2   Faculty of Information Technology, Monash University, Clayton, VIC 3800, Australia
*   Correspondence: elodie.camprasse@gmail.com; Tel.: +61-(0)-497-940-793

**Abstract:** Our transformation of global environments into human-dominated landscapes has important consequences for wildlife. Globally, wildlife is interacting with humans or impacted by human activities, which often results in negative outcomes such as population declines, disruption of social bonds, biodiversity loss, imperilment of threatened species, and harm to individual animals. Human and non-human threats to wildlife can be challenging to quantify and tend to be poorly understood especially over large spatial scales and in urban environments. The extent to which such damage is mitigated by reactive approaches (e.g., wildlife rescue) is also not well understood. We used data from the main state-based Wildlife Emergency Response Services (WERS) in Victoria, Australia to address these issues. The data, which describe tens of thousands of cases of threats to wildlife annually over a ten-year period, allowed a detailed characterisation of the type and extent of threats in the state. We identified the main common and threatened species impacted by various threats and showed that the vast majority of them were anthropogenic (e.g., vehicle collisions, cat attacks, and entanglements). The extent to which different taxonomic groups and species were impacted by various threats differed and threats were dependent on locations. The Greater Melbourne area was identified as a hotspot for threats to wildlife. The WERS was able to source service providers for thousands of animals annually, facilitating their assessment, release into the wild and rehabilitation. However, every year, thousands of animals died or were euthanased and thousands more were left unattended. WERS case reports are increasing and there is a growing service–demand gap. Whilst studies reporting on the demand and response of WERS are rare, situations in other parts of Australia and the world might be similar. This highlights the urgent need to understand and mitigate human and non-human threats to wildlife, particularly in urban environments, where the rate of biodiversity loss is high. We discuss opportunities and barriers to doing so.

**Keywords:** threats to wildlife; urban biodiversity; wildlife rescue; wildlife emergency response services; anthropogenic activities; human impacts; human–wildlife interactions

## 1. Introduction

### 1.1. Human and Non-Human Threats to Wildlife

Our transformation of global environments into human-dominated landscapes has important consequences for non-human animals (hereafter animals). Rapid human population growth and urbanisation can lead to positive impacts for wildlife [1]. Often, however, they result in negative impacts on wildlife, including population declines, disruption of social bonds, biodiversity loss, imperilment of threatened species, and harm to individual animals [2–8]. Globally, wildlife is indeed negatively impacted directly by a range of anthropogenic activities (e.g., vehicle collisions, attacks by domestic pets, entanglements, and gardening incidents) [6–14]. Systematic, large-scale and multi-taxonomic approaches

to understand what these threats are and how wildlife rescue services might be able to respond to these threats are rare.

Human–wildlife interactions pose a real or perceived threat to either party [15]. Human–wildlife interactions have sometimes been referred to as human–wildlife conflicts in the literature [14–17]; though see [18] for a discussion of why that terminology can be misleading and harmful. For humans, negative interactions with wildlife include risks from injury, disease transmission, aggression, and experiences with 'nuisance' animals (e.g., possums in roofs) [19]. Another negative interaction for humans is exposure to distressed, injured or deceased animals. For animals, negative experiences include displacement, harm (physical and psychological trauma), disease transmission and death [6,14,16].

### 1.2. Underutilised Data for Understanding Human and Non-Human Threats to Wildlife

Wildlife Emergency Response Services (WERS) provide rescue, rehabilitation and release for millions of animals worldwide, reducing untold amounts of suffering and death [20]. They include wildlife emergency hotlines as well as rescue and rehabilitation centres. For people witnessing wildlife suffering and perceived or real emergencies, WERS provide education, support and advice [21–23]. WERS collect information from community members concerned about the safety and welfare of wildlife, or less commonly from community members concerned about their own safety as a result of wildlife presence or behaviours. WERS assess the need for a qualified person to attend animals reported to them and coordinate emergency response when needed. To do this, WERS rely on networks of volunteers and other organisations trained to assess, rescue, rehabilitate, and euthanase wildlife. Following assessment from volunteers and/or veterinarians, wildlife is released into the wild if possible or taken into rehabilitation [8,24]. When rehabilitation into the wild is deemed compromised, wildlife is euthanased [9,25,26].

WERS collect temporal, spatial and species-specific human–wildlife interaction data across large regions [20]. WERS data provide an understanding of the types of threats to wildlife, changes in threats through time, and locations of threats [6,11,14,27]. Many operate seven days a week year-round, which results in large and rich databases. These data give insights into the vulnerability of populations relative to their life history and can inform management and conservation of species [28,29]. WERS also provide direct and indirect targeted education that is contextually, temporally, and locally relevant. For example, WERS operators might explain that an uninjured fledgling on the ground facing no visible threats should be left alone rather than "rescued" or relocated. They might also explain the importance of responsible pet ownership to protect wildlife to members of the public unaware of the impacts their pets might have on local species. Provocatively, Tribe and Brown [21] suggested that the greatest benefit derived from WERS is the incidental public education they provide regarding wildlife issues.

Australia is home to a large biodiversity of fauna and flora and ranks high in terms of biodiversity and endemism worldwide [30,31]. However, it ranks very poorly in terms of conservation, with one of the worst records of biodiversity loss globally [32]. Australian urban environments are uniquely positioned to sustain vulnerable and threatened species as cities are hotspots for threatened species, whose range are now sometimes limited to urban environments [33–35]. WERS are present in all Australian states and play a crucial part in reducing harm inflicted on animals from human–wildlife interactions and other non-human threats (e.g., severe weather, predation or interactions with other wildlife) and assisting the community during wildlife emergencies [9,21,28,36]. Australian WERS rely on numerous rescuers and rehabilitation facilities to perform a substantial number of rescues [9,21,36]. Nationally, over 20,000 wildlife carers rehabilitate injured and orphaned wildlife [9]. In New South Wales alone, volunteers have collectively reported over 1,000,000 wildlife rescues over 16 years [37]. Melbourne, Victoria has the largest growth for any Australian capital city and is the most densely populated area in Australia [38], hence the importance of investigating threats to wildlife in this state.

Human–wildlife interactions and other threats can be challenging to quantify and tend to be poorly understood especially in urban environments [14,39]. To date, except for a few Australian examples [6,27], studies utilising WERS data have focused mainly on identifying common causes of morbidity and mortality for specific species or taxa [4,5,40,41] or linked to specific threats [42]. A broader analysis of WERS data considering various taxonomic groups, threat types and spanning long timeframes is recommended to identify human and non-human threats of particular significance for successful wildlife management and conservation [14,24]. In addition, in Australia or elsewhere no published study that we know of has attempted to characterise the demand to WERS services and the extent to which this demand is met.

Here, we analyse a large WERS dataset to characterise the impacts of human activities on wildlife, WERS demand and response. The data describe hundreds of thousands of cases in which wildlife faced human and non-human threats collected over ten years by Wildlife Victoria, a state-based WERS in Victoria, Australia. Specifically, we aimed to: (1) describe the taxonomic groups and species affected, the types of threats reported and associated spatial patterns; (2) determine WERS demand and how it is met through time; and (3) characterise the outcome of reported cases, and WERS' provision of education and service providers. Based on our analyses, we discuss opportunities and barriers to improving our understanding of human and non-human threats and the way they are mitigated.

## 2. Methods

### 2.1. Study Area

Wildlife Victoria, established in 1989, is the main not-for-profit WERS in Victoria, Australia. Victoria is located in South-East Australia and has a population of nearly 6.7 million inhabitants [43]. Melbourne is Victoria's biggest city, with a population of 5159 million inhabitants (~3/4 of the Victorian population); it has the largest growth for any Australian capital city and is the most densely populated area in Australia [38]. Victoria consists of suburbs (i.e., localities whose boundaries are defined by local governments), organised in Local Government Areas (LGAs).

### 2.2. Wildlife Emergency Response Service and Analysed Data

Operators at the Wildlife Victoria's call centre receive information from community members who want to report animals in real or perceived need of support via calls and reports made through an online web-portal or a mobile phone app. The opening hours of the call centre varied between years throughout the study period and sometimes between seasons (longer operating hours in spring and summer than in fall and winter to respond to a higher volume of calls). However, the call centre was usually open seven days (excluding some public holidays at times), between 6:30 a.m. and 8:30 p.m. Wildlife Victoria operators are trained, employed professionals (though at times, trained volunteers stepped in to man the call centre during public holidays). A single case report can be made for a single individual or a group of animals (parents and siblings, mobs, flocks, etc.).

Wildlife Victoria relies on an extensive state-wide network of volunteers to rescue and rehabilitate native animals in need, as well as veterinarians who assess wildlife pro-bono. We used the reports made on a daily basis to Wildlife Victoria from 1 January 2010 to 31 December 2019 to determine how human and non-human threats affect different wildlife species in Victoria. The dataset consists of temporal and spatial information on those threats; each case provides information on the species, the reason for the report, who responded to the case and the final fate (Table 1). Whilst some cases referred to Wildlife Victoria require rehabilitation, the majority of them do not. As such, we focused on what can be learnt about human and non-human threats to wildlife through the rescue process, more so than through the rehabilitation process.

**Table 1.** Structure of Wildlife Victoria's dataset (cases reported to the Emergency Response Service at Wildlife Victoria between 2010 and 2019) used for analysis.

| Case Data | Details |
| --- | --- |
| Date and time | dd/mm/yyyy HH of the report |
| Suburb where the animal(s) were reported | Describes the Victorian suburb where the animal(s) were reported |
| Case number | Unique ID of a case |
| Species or group of species name | See Supplementary Material Table S1 for all species and groups of species |
| Cause type (i.e., reason why the animal(s) were reported) | Cause type from any of the 21 identified categories:<br>● hit by vehicle;<br>● abnormal location—where the animal(s) were reported in an unnatural habitat where they did not belong and where their safety was assumed to be compromised, e.g., in carparks and busy streets;<br>● entangled—in fence, netting or rubbish;<br>● found within building;<br>● trapped—where animal(s) could not get out on their own of an enclosed space, such as pipes, drains, etc.;<br>● attack by cat;<br>● attack by dog;<br>● attack by other animal (e.g., bird and fox);<br>● collision—where the animals collided with a fixed human-made structure such as a buildings, windows, poles or power lines;<br>● nuisance—where members of the public wanted to complain about perceived negative impacts animal(s) were causing and/or wanted the animal(s) relocated because of such impacts;<br>● orphaned—where a young animal was found without its parents;<br>● disease;<br>● extreme weather;<br>● pet—where the members of the public called about an animal kept as a pet<br>● cruelty—where the animal(s) reported were victims of act of cruelty;<br>● fledging—where a bird was found without its parents;<br>● escapee—where a captive animal(s) escaped from where they were kept such as rehabilitation facilities, farms, etc.;<br>● poisoned;<br>● oil spill;<br>● found on the ground—where the animal(s) were found on the ground for an unknown reason;<br>● unknown. |
| The final fate (i.e., outcomes of the report) | Final fate from any of the nine identified categories:<br>● connected to another case;<br>● euthanized—when the recovery of the animal(s) was deemed too compromised to successfully rehabilitate them into the wild;<br>● died—when the animal(s) died as a result of their injuries or condition before or after being assessed and/or rescued;<br>● in rehabilitation—when the animal(s) were taken to a foster carer for ongoing treatment and care with the aim of releasing them into the wild;<br>● advice/education given—when the member of the public received advice or education regarding a case Wildlife Victoria could not respond to (e.g., non-native and invertebrates) or a case where the animal(s) needed to be left alone;<br>● no rescue required—when the emergency response operator determined that no action was required from Wildlife Victoria's volunteers or from the member of the public;<br>● referred to other organisation—when the member of the public was asked to contact a commercial service for animal removal or when the case was passed on to organisations such as Zoos Victoria, the Marine Response Unit, the Department of Environment, Land, Water and Planning;<br>● released to the wild—when the animal(s) were assessed and/or rescued and subsequently released to the wild as they were deemed fit enough;<br>● unknown. |

**Table 1.** *Cont.*

| Case Data | Details |
|---|---|
| The numbers and types of service providers (i.e., who responded to the case to assess, rescue and/or rehabilitate the animal(s) reported) | Service provider type from any of the following eight categories: <br>● rescuer—a Wildlife Victoria volunteer who has the skills and ability to assess and rescue wildlife; <br>● transporter—a Wildlife Victoria volunteer who does not necessarily have animal handling experience but can transport contained animals, e.g., from a veterinarian to a carer; <br>● carer—a Wildlife Victoria volunteer who is licensed to rehabilitate wildlife either as a shelter or an individual carer; <br>● species or disease advisor—a person who has expertise with a particular species or disease and can offer advice about the case; <br>● veterinarian—a veterinarian clinic or practice, which is required to assess wildlife brought to them as a pro bono service; <br>● Police—police officers authorised to dispatch severely injured wildlife (e.g., kangaroos and wombats); <br>● external wildlife service provider—an organisation that is not affiliated with Wildlife Victoria but can send people qualified to assess or rescue animals, e.g., Parks Victoria rangers and government wildlife officers; <br>● unknown. |

The data were extracted through Salesforce, the web-interface Wildlife Victoria uses to record cases. Species names, when known, were associated with their taxonomic group (bird, mammal, reptile, amphibian, invertebrate, unknown), status (native vs. non-native), latin names and conservation status; using databases including the Victorian Biodiversity Atlas, the Atlas of Living Australia and the Department of Environment, Land, Water and Planning Flora and Fauna Guarantee Act 1988 Threatened List [44].

We excluded reports for pets (0.9% of cases), and duplicates (i.e., any subsequent reports after a case was first logged, 3.3% of cases) (Figure 1). After giving an overview of the data (Supplementary Material Tables S1 and S2), we excluded reports for invertebrates (0.1% of cases), and some marine species (all marine species excluding birds, 0.8% of cases) because Wildlife Victoria cannot send volunteers to these cases. For an overview of all cases, we included non-native species (7.3% of cases) to capture issues related to all wildlife species that were likely to be present and reported to the WERS. Only when analysing service provider data (i.e., whether or not a service provider, e.g., a rescuer, was found to respond to a specific case), we excluded cases for non-native wildlife as Wildlife Victoria cannot send volunteers to non-native species.

Cases that require action (i.e., the animal(s) need to be assessed, rescued and/or rehabilitated) are cases for native species (amphibians, birds, mammals, reptiles, excluding non-native species, invertebrate and "unknown" taxonomic group, pets, marine mammals, marine reptiles and marine fishes). Cases that require action also exclude cases for which operators determine that no rescue is required or give advice/education to the community member (Figure 1). When WERS operators determine that a case needs to be actioned, they contact volunteers; they include Wildlife Victoria's transporters (volunteers who transports wildlife; e.g., from veterinary practices to carers or vice versa), rescuers, carers, or professional service providers (e.g., police and park rangers) depending on the situation.

A case can have several service providers linked to it (e.g., a case for which a rescuer retrieves the animal and takes it to a veterinarian, where the animal is assessed, and then handed over to a carer; such case would have three linked service providers). A single service provider can be linked to a case multiple times under a different service provider type (e.g., the same person can be linked first as a rescuer, and then linked as a carer to reflect their specific role in the process).

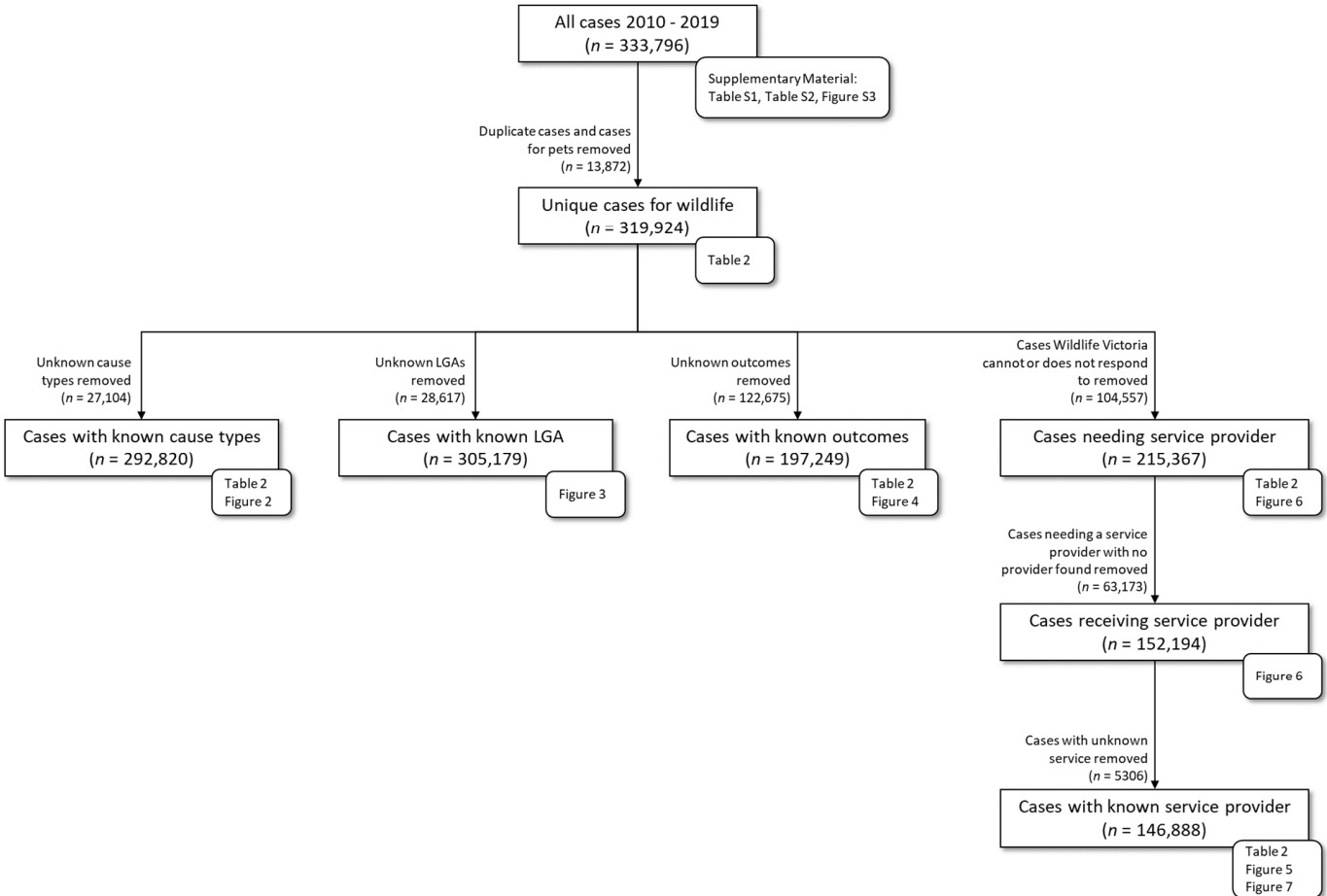

**Figure 1.** Dataset available and sample sizes for different subsets of the data; cases reported to Wildlife Victoria between 2010 and 2019.

### 2.3. Data Handling and Analysis

The Salesforce dataset was cleaned, corrected, summarised and added to an SQLite database. To create the plots, we created a web-application using the Plotly library [45]. Data were visualised using choropleth maps with cases mapped to LGAs using the relevant LGA boundary information [46]. LGAs were determined as being part of the Greater Melbourne area or the rest of the state [47]. Population growth for the state of Victoria from 2010 to 2020 was obtained [38]. Summaries presented are means ± standard deviations.

Chi-square goodness of fit tests (thereafter chi-square tests) were performed to check differences between observed and expected case numbers after checking that the right assumptions were met (random sampling, using counts of cases rather than percentages, making sure all expected counts equal to five or greater). Expected values for the chi-square tests (except for the flying fox case study) for each group are determined by the following formula: (column total X row total)/overall total in contingency tables comprising the numbers of cases for each category. Chi-square tests were performed on top ten species and top ten causes, rather than all species and causes. For the flying fox data, we expected that 71.7% of cases would be within Greater Melbourne (as indicated for the rest of the dataset). To avoid excpected values of less than five, we narrowed down the number of categories from ten to five, eliminating categories with small sample sizes but still keeping more than 90% of flying foxe cases. Main differences are presented in the results (i.e., groups which have the highest differences from expectations). Those tests were conducted in the R Statistiscal Environment in version 4.2.1 [48]. All data were tested for significance at the 5% level.

## 3. Results

### 3.1. WERS Data Overview

From 1 January 2010 to 31 December 2019, 333,796 cases were reported to the WERS. Duplicate cases and cases for animals kept as pets were removed ($n$ = 13,872), which resulted in the retention of 319,924 unique cases for wildlife, with a mean of 31,992 ± 8783 cases annually (~88 ± 39 cases a day; min = 7, max = 339).

#### 3.1.1. Overview of Species Affected

A total of 573 different species or groups of species were represented in the dataset (Supplementary Material Table S1). Of these, 443 were actual, identified species (i.e., excluding unidentified groups of species such as "bird unidentified", "possum unidentified", etc.); 81 species were Flora and Fauna Guarantee Act 1988 (FFG) listed (thereafter referred to as threatened species) and reported in the whole database (both outside and inside of the Greater Melbourne area, $n$ = 4080 cases). Furthermore, 63 species were threatened and reported in the Greater Melbourne area only ($n$ = 3024 cases; Supplementary Material Tables S1 and S2). Overall, there were 408 ± 93 cases annually for threatened species in Victoria, including 302 ± 75 in Greater Melbourne. Of the threatened species in Greater Melbourne, the majority were grey-headed flying foxes *Pteropus poliocephalus* (hereafter, flying foxes, $n$ = 2588, 89.8% of cases for Greater Melbourne threatened species).

Mammals and birds dominated cases reported to the WERS (96.0% of cases), and the rest comprised of reptiles, amphibians, invertebrates and unknown species (Table 2). Most cases were logged for species that were not FFG listed (hereafter referred to as common species, $n$ = 319,924, 98.7% of cases) (Table 2). The top ten species or group of species reported (58.0% of all cases) were: the eastern grey kangaroo (*Macropus giganteus*—hereafter referred to as kangaroo); the common ringtail possum (*Pseudocheirus peregrinus*—hereafter referred to as ringtail); the Australian magpie (*Gymnorhina tibicen*—hereafter referred to as magpie); the common brushtail possum (*Trichosurus vulpecula*—hereafter referred to as brushtail); unidentified bird species; the rainbow lorikeet (*Trichoglossus moluccanus*); unidentified possum species; the raven (*Corvus coronoides*); the bare-nosed wombat (*Vombatus ursinus*—hereafter referred to as wombat); and the koala (*Phascolarctos cinereus*) (Table 2).

**Table 2.** Total number of unique cases, percentage of total unique cases, percentage of unique cases with known fate, number of unique cases to action with a known service provider, percentage of unique cases to action with a known service provider, leading known cause and leading known outcome, broken down by taxonomic group, species or group of species, native status and listing under the Flora and Fauna Guarantee (FFG Listed) Act 1988 Threatened List for cases reported to the Emergency Response Service at Wildlife Victoria between 2010 and 2019.

| | Total Number of Unique Cases | Percentage of Total Unique Cases (%) | Percentage of Unique Cases with Known Fate (%) | Number of Unique Cases to Action/Number of Cases to Action with a Known Service Provider | Percentage of Unique Cases to Action with a Known Service Provider (%) | Leading Known Cause | Leading Known Outcome |
|---|---|---|---|---|---|---|---|
| **Taxonomic group** | | | | | | | |
| Mammal | 163,840 | 51.2 | 60.9 | 132,413/93,669 | 70.7 | Hit by vehicle (44.6% of cases) | Euthanased (24.3% of cases) |
| Bird | 143,418 | 44.8 | 61.3 | 78,422/50,579 | 64.5 | Hit by vehicle (21.5% of cases) | Advice/education given (48.0% of cases) |
| Reptile | 11,322 | 3.5 | 74.2 | 4560/2629 | 57.7 | Abnormal location (24.1% of cases) | Advice/education given (53.1% of cases) |
| Unknown | 567 | 0.2 | 77.4 | NA | NA | Found within building (28.4% of cases) | Advice/education given (72.0% of cases) |

**Table 2.** *Cont.*

| | Total Number of Unique Cases | Percentage of Total Unique Cases (%) | Percentage of Unique Cases with Known Fate (%) | Number of Unique Cases to Action/Number of Cases to Action with a Known Service Provider | Percentage of Unique Cases to Action with a Known Service Provider (%) | Leading Known Cause | Leading Known Outcome |
|---|---|---|---|---|---|---|---|
| Amphibian | 540 | 0.2 | 81.5 | 43/20 | 46.5 | Displaced native (67.1% of cases) | Advice/education given (50.9% of cases) |
| Invertebrate | 341 | 0.1 | 87.7 | NA | NA | Nuisance (47.4% of cases) | Advice/education given (58.2% of cases) |
| Native status | | | | | | | |
| Native | 269,682 | 84.3 | 58.8 | 215,438/146,897 | 68.2 | Hit by vehicle (38.3% of cases) | Advice/education given (25.3% of cases) |
| Non-native | 23,297 | 7.3 | 91.3 | NA | NA | Attack by cat (16.1% of cases) | Advice/education given (82.5% of cases) |
| Unknown | 27,049 | 8.5 | 64.3 | NA | NA | Abnormal location (15.9% of cases) | Advice/education given (60.1% of cases) |
| FFG Listed | | | | | | | |
| No | 315,945 | 98.7 | 60.8 | 232,048/152,140 | 65.6 | Hit by vehicle (35.5% of cases | Advice/education given (34.7% of cases) |
| Yes | 4083 | 1.3 | 68.1 | 3441/2743 | 79.7 | Entangled (48.4% of cases) | In rehabilitation (29.8% of cases) |
| Species or group of species | | | | | | | |
| Eastern grey kangaroo *Macropus giganteus* | 49,188 | 15.4 | 61.1 | 44,940/31,428 | 69.9 | Hit by vehicle (72.1% of cases) | Euthanased (46.0% of cases) |
| Common ringtail possum *Pseudocheirus peregrinus* | 48,296 | 15.1 | 53.7 | 43,578/32,617 | 74.8 | Attack by cat (21.3% of cases) | In rehabilitation (41.5% of cases) |
| Australian magpie *Gymnorhina tibicen* | 19,189 | 6.0 | 55.0 | 13,473/7980 | 59.2 | Hit by vehicle (24.1% of cases) | Advice/education given (44.4% of cases) |
| Common brushtail possum *Trichosurus vulpecula* | 19,089 | 6.0 | 62.6 | 15,528/10,874 | 70.0 | Found withing building (23.8% of cases) | In rehabilitation (26.6% of cases) |
| Bird, unidentified | 14,012 | 4.4 | 63.4 | NA | NA | Found withing building (22.9% of cases) | Advice/education given (66.4% of cases) |
| Rainbow lorikeet *Trichoglossus moluccanus* | 10,456 | 3.3 | 56.2 | 8449/6107 | 72.3 | Collision (26.7% of cases) | In rehabilitation (32.6% of cases) |
| Possum, unidentified | 7819 | 2.4 | 55.3 | 5728/2856 | 49.9 | Found within building (28.6% of cases) | Advice/education given (41.4% of cases) |
| Raven *Corvus coronoides* | 5961 | 1.9 | 52.9 | 4261/2464 | 57.8 | Hit by vehicle (28.2% of cases) | Advice/education given (44.0% of cases) |

| | Total Number of Unique Cases | Percentage of Total Unique Cases (%) | Percentage of Unique Cases with Known Fate (%) | Number of Unique Cases to Action/Number of Cases to Action with a Known Service Provider | Percentage of Unique Cases to Action with a Known Service Provider (%) | Leading Known Cause | Leading Known Outcome |
|---|---|---|---|---|---|---|---|
| Bare-nosed wombat *Vombatus ursinus* | 5954 | 1.9 | 59.4 | 4813/3217 | 66.8 | Hit by vehicle (45.8% of cases) | Advice/education given (23.6% of cases) |
| Koala *Phascolarctos cinereus* | 5914 | 1.8 | 57.3 | 4485/3333 | 74.3 | Hit by vehicle (37.1% of cases) | No rescue required (21.5% of cases) |

### 3.1.2. Cause Types

In 56.4% of cases, the cause type of the report could not be determined. The top ten cause types reported for cases with known causes (91.5% of cases with known causes) were: hit by a vehicle (*n* = 48,800, 35.0%); abnormal location (*n* = 17,926, 12.9%); entangled (*n* = 11,186, 8.0%); found within building (*n* = 10,946, 7.8%); attacked by cat (*n* = 8084, 5.8%); trapped (*n* = 7298, 5.2%); attacked by dog (*n* = 6456, 4.6%); attacked by other animals (*n* = 6340, 4.5%); victims of collisions (*n* = 5366, 3.8%); and cases considered nuisance (*n* = 5346, 3.8%).

The breakdown of cause types varied between species, both in relation to variety and proportion (Figure 2, Table 2). For example, the most frequent threats, hit by a vehicle, affected kangaroos and wombats more than any other species or group of species; entanglement was a cause of concern for threatened species, with more than half of cases for threatened species resulting from this cause type (Figure 2, Table 2).

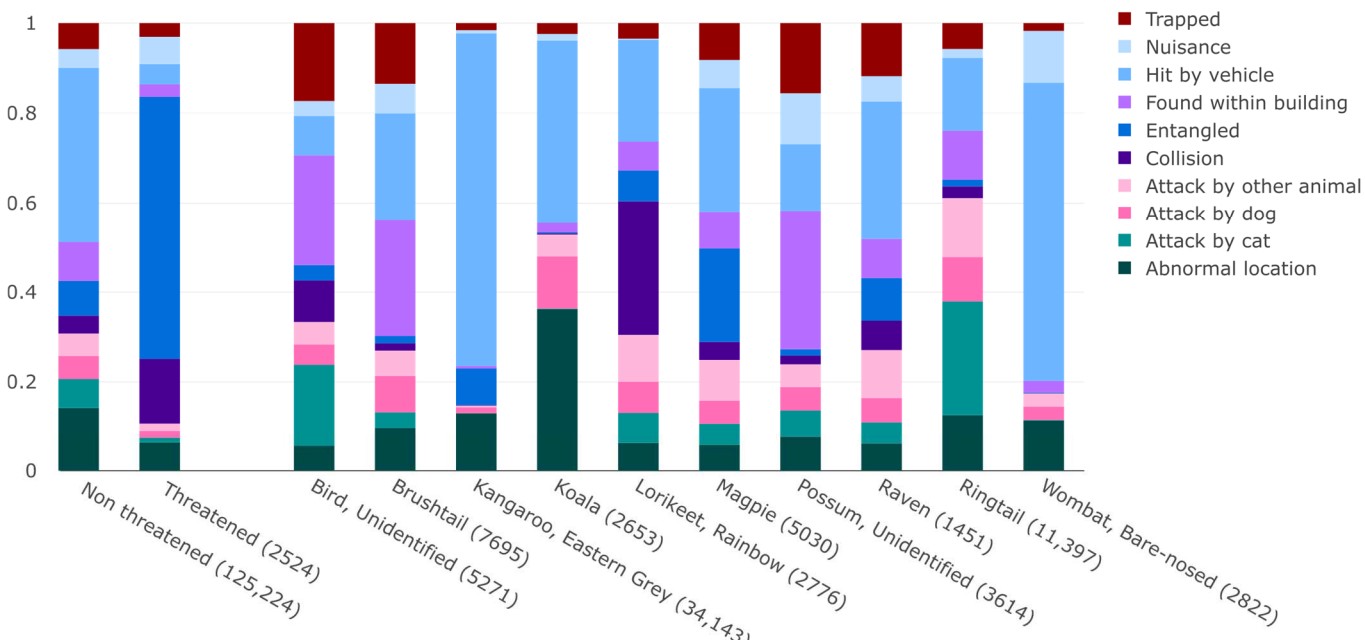

**Figure 2.** Break down of known top ten cause types (excluding cases with cause type unknown and found on the ground) for cases reported to Wildlife Victoria between 2010 and 2019 with sample sizes for each category; top ten causes for wildlife that is non-threatened and threatened in accordance with the Flora and Fauna Guarantee Act (FFG) 1988 (left hand side) and top ten causes for top ten species (right hand side) (*n* = 127,748).

### 3.1.3. Spatial Patterns

Cases were assigned to an identified suburb when possible and then an LGA. Some "suburbs" present in the database did not match any currently known Victorian suburbs and LGA, and were removed from further analyses. In total, 305,179 cases (95% of all unique cases that excluded duplicates and cases made for pets) were successfully assigned to an LGA and subsequently identified as being part of the Greater Melbourne area or not. Out of cases with a known LGA, 71.8% ($n$ = 219,090) were located in Greater Melbourne, despite this area representing only approximately 4.4% of the surface of Victoria [49,50] (Figure 3, Supplementary Material Table S2).

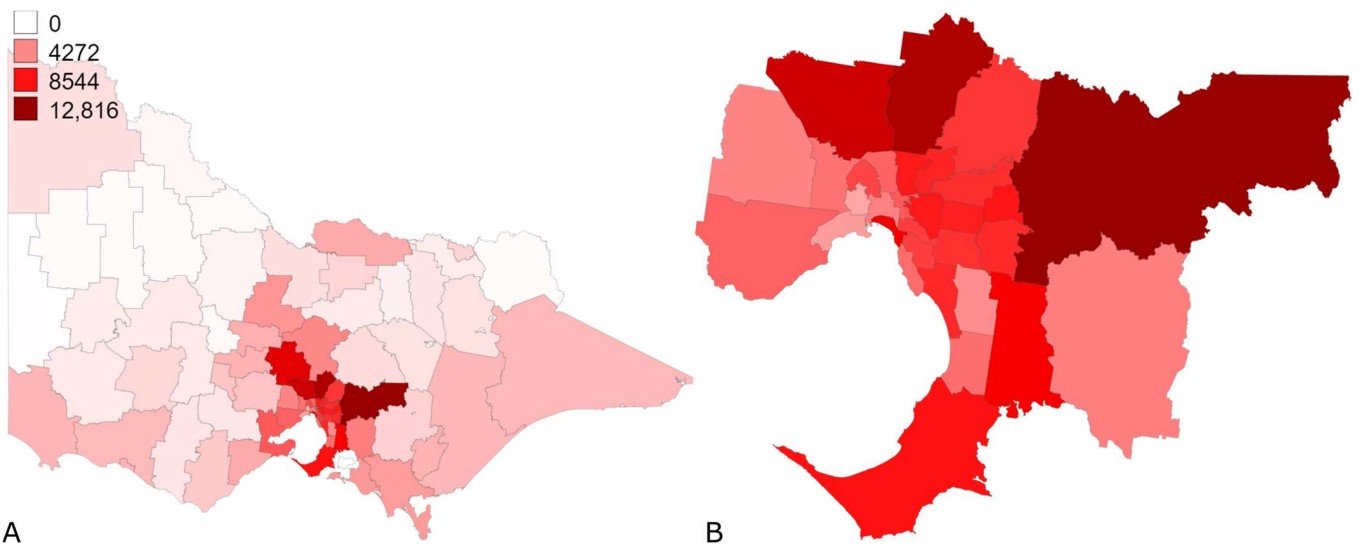

**Figure 3.** Maps indicating the number of cases in different LGAs for cases reported to Wildlife Victoria between 2010 and 2019 in (**A**) the state of Victoria ($n$ = 305,157); (**B**) the Greater Melbourne area ($n$ = 219,068).

There were statistical differences in the number of case numbers reported for wildlife from different taxonomic groups in the Greater Melbourne area compared to the rest of the state ($\chi^2_5$ = 1714.6, $p$ < 0.0001). Birds were significantly more impacted (i.e., more cases were logged) in Greater Melbourne, whilst mammals were more impacted in the rest of the state. The top ten species were also impacted differently depending on whether or not they were in the Greater Melbourne area ($\chi^2_9$ = 44,266.0, $p$ < 0.0001); specifically, kangaroos and koalas were less impacted, and ringtails were more impacted in Greater Melbourne. The main differences between the Greater Melbourne area and the rest of the state in terms of cause types was a higher number of cases for cat attacks in the former and a higher number of vehicle collisions in the latter ($\chi^2_9$ = 9907.4, $p$ < 0.0001).

The top five causes for flying foxes (the majority of threatened species) were: entanglement (63.5% of all cases with known cause types, $n$ = 1377), victim of collisions (15.4% of cases with known cause types, $n$ = 335 cases), considered as nuisance (5.3% of cases with known cause types, $n$ = 114 cases), found in abnormal location (4.2% of cases with known cause types, $n$ = 91 cases) and trapped (3.2% of species with known cause types, $n$ = 70 cases). Combined, these top five causes encompass 91.6% of all flying fox cases. Flying foxes were more frequently reported in Greater Melbourne compared to the rest of the state ($\chi^2_1$ = 178.1, $p$ < 0.0001). However, there were no significant differences in cause types for flying foxes inside vs. outside Greater Melbourne ($\chi^2_4$ = 8.8, $p$ = 0.07)).

### 3.2. WERS Support Services' Current Response to Human–Wildlife Interactions and Demand

### 3.2.1. Outcomes of Cases

Out of all unique cases, 38.3% had unknown outcomes ($n$ = 122,675), 21.3% resulted in giving advice/information to the member of the public ($n$ = 68,151), and it was determined

in 4.5% of cases that no rescue was required (*n* = 14,250). Out of the cases where an intervention was required (*n* = 215,367), 51.3% of cases had known outcomes (*n* = 110,514). Overall, 21,537 ± 5649 cases required an intervention annually (~59 ± 24 cases a day; min = 5, max = 208).

Out of the cases with known outcomes, 33.7% went to a rehabilitation facility (*n* = 35,298), 30.5% were euthanased (*n* = 31,938), 18.3% died (before the service provider arrived or during the rescue, *n* = 19,214), 9.9% were assessed and released back into the wild (*n* = 10,379). The outcomes for the top ten species or group of species were known on average in 57.9% of cases (range = 52.9–63.4%, Table 2).

Seven known outcomes were identified: advice/education given (*n* = 68,153, 34.5% of known outcome cases), in rehabilitation (*n* = 37,029, 18.8%), euthanased (*n* = 33,542, 17.0%), died (*n* = 20,344, 10.3%), no rescue required (*n* = 14,257, 7.2%), referred to other organisation (*n* = 12,829, 6.5%), released to the wild (*n* = 11,136, 5.6%) (Figure 4). There were significant differences between the outcomes of the cases based on taxonomic group ($\chi^2_{30}$ = 23,090, *p* < 0.0001). Overall, bird cases resulted in more and mammal cases resulted in more advice/education given. Bird cases resulted in less, and mammal cases resulted in more, deaths. Bird cases resulted in less, and mammal cases resulted in more, euthanasia. There were significant differences between the outcomes of the cases based on species for the top ten species ($\chi^2_{54}$ = 40,767, *p* < 0.0001). Cases for unidentified birds were more frequently, and cases for mammals were less frequently given advice/education. Cases for kangaroos were more frequently euthanased. Cases for ringtails were more frequently taken into rehabilitation. There were significant differences between the outcomes of the cases based on causes for the top ten cause types and species ($\chi^2_{54}$ = 27,647.0, *p* < 0.0001). Specifically, cases for animals hit by vehicles resulted in significantly more death or euthanasia. Cases for animals found within buildings or reported as nuisance were significantly more likely to receive advice/education. Cases for flying foxes resulted in more rehabilitation, and less "no rescue required" outcomes than expected by chance ($\chi^2_6$ = 774.3, *p* < 0.0001). The main differences between the Greater Melbourne area and the rest of the state in terms of case outcomes was a prevalence for the provision of education and advice in the former and more animals euthanased in the latter ($\chi^2_7$ = 3824.3, *p* < 0.0001).

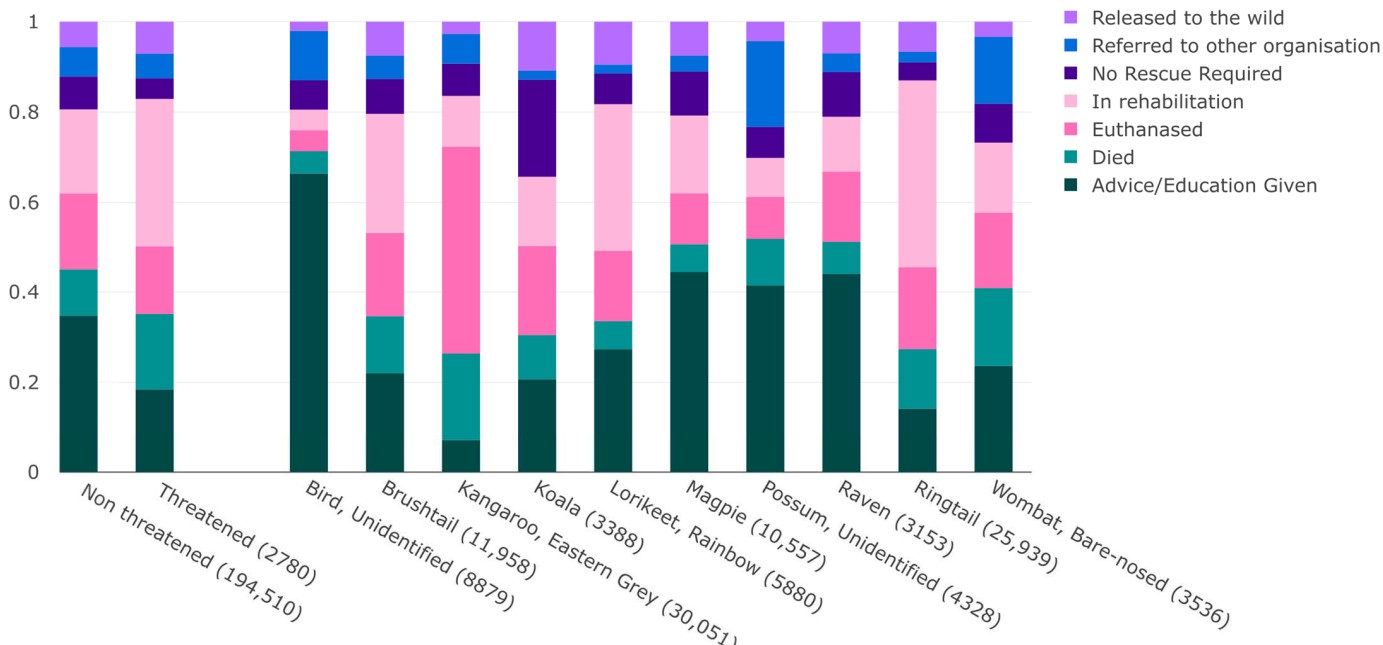

**Figure 4.** Breakdown of known outcomes for cases reported to Wildlife Victoria between 2010 and 2019 with sample sizes for each category; top ten causes for wildlife that is non-threatened and threatened in accordance with the Flora and Fauna Guarantee Act (FFG) 1988 (left hand side) and top ten causes for top ten species (right hand side) (*n* = 197,290).

3.2.2. WERS Demand

The number of cases reported to the WERS increased steadily, from 16,270 unique cases in 2010 to 42,292 cases in 2019, representing a 2.5-fold increase mirroring that of population growth in Vfabnorictoria for the same period (Supplementary Material Figure S1).

The total number of Wildlife Victoria volunteers (transporters, carers, rescuers) peaked in 2013 (*n* = 1466 different volunteers linked to cases), was at its lowest in 2015 (*n* = 905) and increased back up to numbers similar to 2010 (*n* = 1120 volunteers). The number of veterinarians linked to Wildlife Victoria cases have increased between 2010 (*n* = 243) and 2019 (*n* = 492).

Overall, the number of cases for which Wildlife Victoria volunteers were linked to increased from 2010 to 2019, which was mostly driven by the number of cases rescuers were linked to (Figure 5). The number of cases for which veterinarians were linked to followed a similar trend, whilst the number of cases for which police was linked to remained more constant through the ten-year period (Figure 5). The number of cases needing a service provider increased through time, whilst the number of service providers linked to cases reached a plateau and then decreased after 2013. This resulted in a decreasing percentage of cases in need of a service provider and for which the WERS was able to find one (Figure 6).

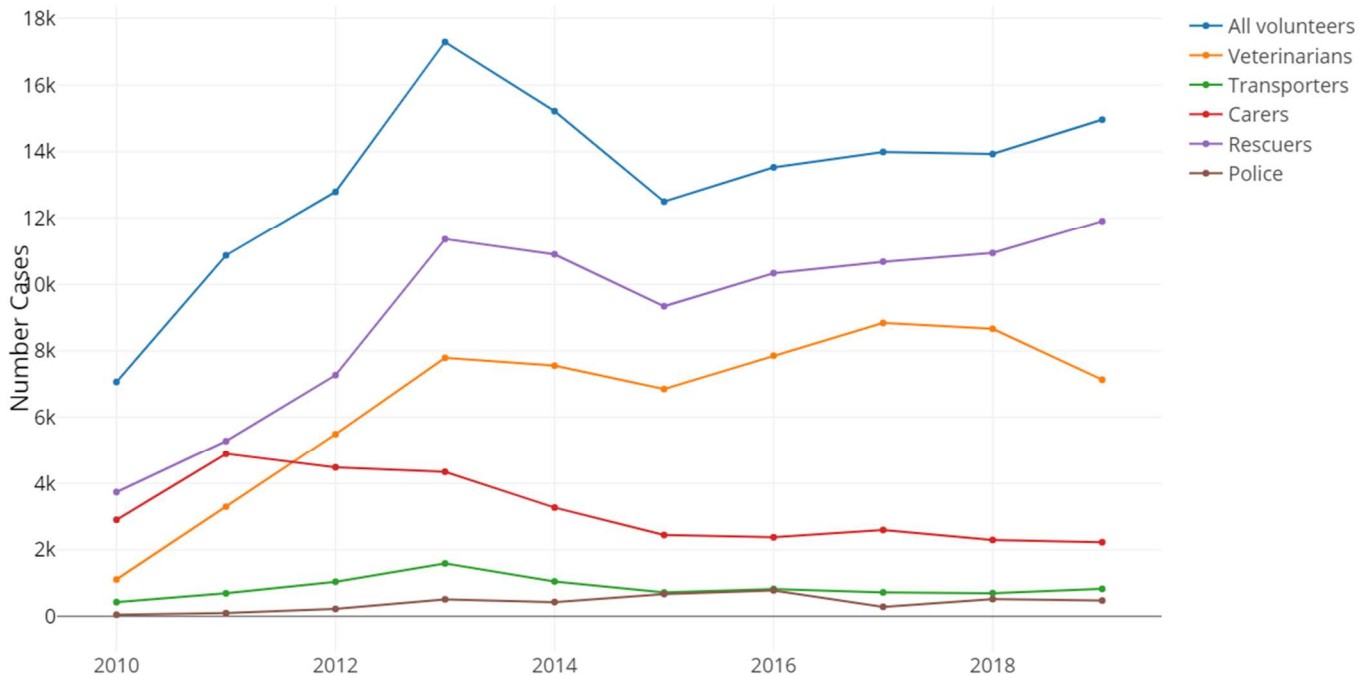

**Figure 5.** Number of cases for which different categories of service providers (all Wildlife Victoria volunteers, all Wildlife Victoria rescuers, all Wildlife Victoria carers, all Wildlife Victoria transporters and all veterinarians) were linked to for cases reported to Wildlife Victoria between 2010 and 2019 (total cases, *n* = 146,897).

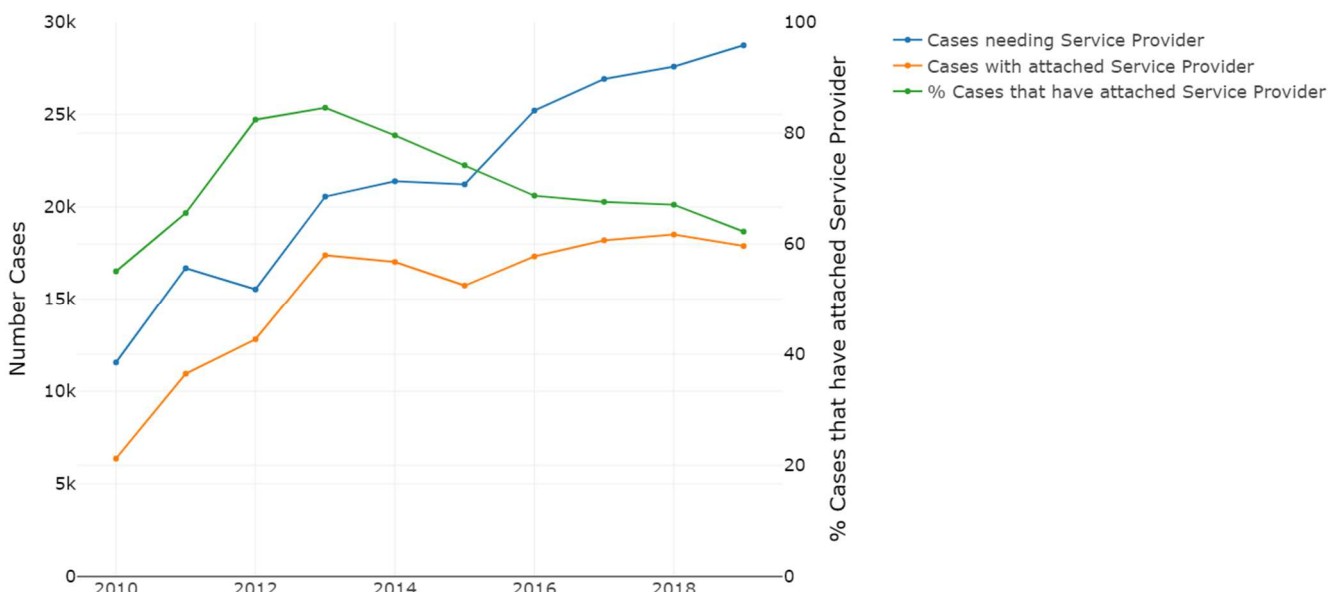

**Figure 6.** Number of cases needing a service provider (i.e., an intervention was required, *n* = 215,367), number of service providers linked to cases (*n* = 152,205), and percentage of cases needing a service provider and obtaining one for cases reported to Wildlife Victoria between 2010 and 2019.

3.2.3. Provision of Education and Service Providers

Advice or education was provided to members of the public reporting a case in 21.3% of all unique cases (*n* = 68,151 cases). Advice/education given was an outcome which was more frequent in common compared to threatened species, in birds (unidentified, magpies, ravens), as well as unidentified possums (Figure 7).

Of the 215,367 cases for which the animal(s) needed to be assessed, rescued and/or rehabilitated: 29.3% (*n* = 63,173) could not get a service provider and were removed from further analysis (~6317 ± 2694 cases per year); 2.5% (*n* = 5280) had a linked service provider but its type was unknown; 68.2% (*n* = 146,914) received an identified service provider.

Service providers were disproportionally found outside of the Greater Melbourne area ($\chi^2_1$ = 763.2, *p* < 0.0001). Some species in need were more likely to get a service provider (e.g., ringtails and flying foxes) compared to others (e.g., unidentified possums, magpies and ravens; $\chi^2_9$ = 3125.6, *p* < 0.0001). Cases associated with different cause types differed in their likelihood of obtaining a service provider: less likely for nuisance, animals found within building, and in abnormal location and more likely for cases where animals were hit by vehicle, victim of entanglement and attacked by cat ($\chi^2_9$ = 1959.6, *p* < 0.0001). Overall, service providers were found in a higher proportion of cases for threatened species compared to common species ($\chi^2_1$ = 411.9, *p* < 0.0001).

Cases with at least one linked service provider had up to five different linked providers and on average 1.2 ± 0.5 total provider per case. Cases for possums, rainbow lorikeets, magpies and ravens were cases that most often had a linked veterinarian; veterinarians were linked to cases in higher proportions for common species compared to threatened species (Figure 7). Veterinarians were relied upon for some cause types more than others (e.g., attack by cat, dog or other animal, and collision); the majority of cases where the police was linked were cases for kangaroos and cases where the animal(s) were hit by vehicle (Figure 7). Cases with the highest proportion of linked carers were cases for ringtail and brushtail possums, rainbow lorikeets and magpies.

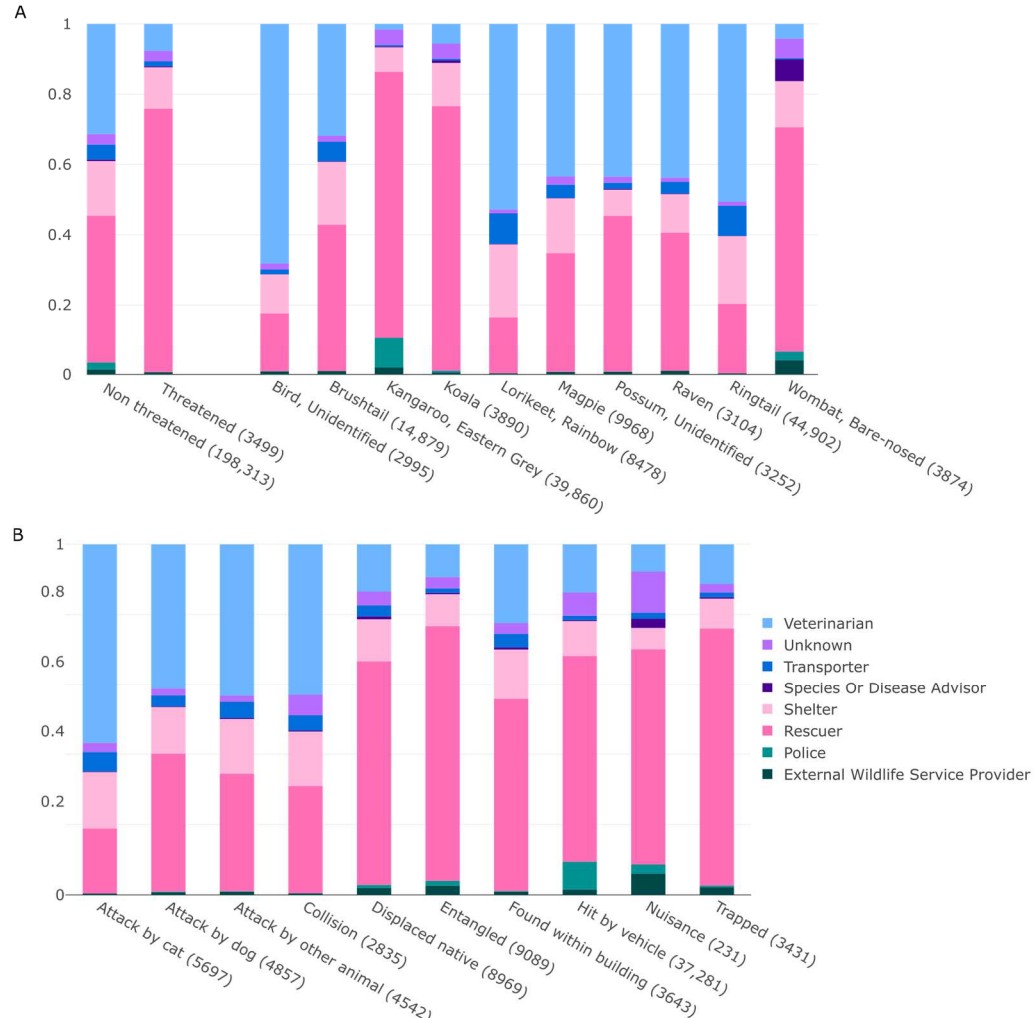

**Figure 7.** Break down of service providers found for cases reported to Wildlife Victoria between 2010 and 2019 and needing an intervention (assessment, rescue and/or rehabilitation) with sample sizes for each category: (**A**) for wildlife that is non-threatened and threatened in accordance with the Flora and Fauna Guarantee Act (FFG) 1988 (left hand side) and top ten causes for top ten species (right hand side) (*n* = 201,812); (**B**) for top ten causes (*n* = 80,575).

## 4. Discussion

Our study demonstrates the value the main WERS in Victoria, Australia provides; it responds to tens of thousands of cases of human and non-human threats to wildlife per year and educates the public on wildlife issues. It also keeps records allowing the characterisation of threats to wildlife, the demand to WERS services and the way wildlife emergencies are attended to. The main reasons why wildlife was reported to the WERS were anthropogenic. Furthermore, different taxonomic groups and different species were impacted differently by human activities. Hotspots of human–wildlife interactions occurred in the Greater Melbourne area, a growing metropolitan area. Worryingly, our results show a 2.5-fold increase in cases over a ten-year period, and a growth in the number of cases that do not receive WERS support despite requiring it, highlighting a growing service–demand gap. The outcomes of cases and the likelihood of finding a service provider for a case depended on the location of the case, the species affected and the reasons why cases were reported. Below, we discuss the insights gained from WERS data in relation to human and non-human threats to wildlife and their mitigation. We consider WERS demand and response, before mentioning the limitations of this study. Lastly, we summarise the

implications of this study and make recommendations on how to better understand and mitigate threats to wildlife.

### 4.1. Impacts of Anthropogenic Activities on Wildlife and Their Mitigation

Wildlife Victoria supports tens of thousands of community members and animals every year, in line with what has been reported for WERS in Australia and internationally [14,20,21,25,26,37,41]. WERS case reports provide a rich and detailed dataset for helping us understand human and non-human threats to wildlife, benefitting from a constant collection of information from public reports on incidents including species, location, date, incident cause type, and outcome [16].

Most species reported to the WERS were common, as per expected [6,7,24]. However, even common species can be subjected to local declines (e.g., kangaroos in Australia [6]) with potential long-term consequences. The threats participating to these declines need to be taken into account to inform their management [39,41,51,52]. In addition, it has been shown that the loss or reduction in some common species can lead to substantial changes in patterns of biodiversity [53].

We identified the main species affected and the main types of human and non-human threats to wildlife over a ten-year period in Victoria. Overall, mammals and birds were by far the most represented taxonomic groups in this dataset, as in other WERS studies, possibly linked to their dominance, particularly in the urban environment [6–8,14,24,54,55]. A few hundred cases for threatened species (with around 3/4 in Greater Melbourne) were reported to the WERS annually; most of these were flying foxes. This is likely explained by the fact that unlike other threatened species, populations of this species in urban areas such as Melbourne comprise of tens of thousands individuals roosting together [5].

We identified the main (i.e., top ten) threat to wildlife. Though non-human threats such as severe weather and interactions with other animals were reported, we showed that threats to wildlife were caused mostly by anthropogenic activities. The threats included a range of factors, from physical aspects of the built environment, people's behaviours (e.g., letting their pets roam free leading to cat and dog attacks) to perceptual issues (e.g., people unsure of animals' place in cities leading to nuisance reports). For example, studies internationally and in Australia have analysed wildlife rehabilitation centre and emergency hotline data for various taxonomic groups and species and determined that car collisions, collisions with fixed structures, and attacks by pets rank high amongst the reason why wildlife is typically admitted [6,13,14,16,24,55–58]. This indicates that these patterns are likely to be global.

Human–wildlife interactions impact specific species or groups of species differently. For example, in our dataset, kangaroos and wombats were frequently reported as victims of vehicle collisions. Taylor-Brown et al. [6] similarly showed that kangaroos were often taken into rehabilitation because of car collisions. The disproportionate effect of some threats over others for different species is not unexpected as it reflects the species ecology and habitat [11,55,59], but it highlights that threat mitigation will vary for each species of concern and needs to be evidence-based.

The Greater Melbourne area encompasses a sprawling urban matrix of almost 10,000 km$^2$ and represents a hotspot for WERS reports. This is unsurprising given that those reports come from the public and locations with higher density have greater potential for human-animal interactions. However, given the magnitude of urbanisation throughout the globe, this issue is likely to apply to other urbanised areas internationally [55]. The high number of case reports in the Greater Melbourne area may also reflect urbanites' expectations to access support in wildlife interactions [17,23]. By identifying hotspots of human–wildlife interactions for threatened species and identifying the most frequent threat types by species and location, WERS data could inform targeted management strategies.

Recent research indeed champions the use of WERS data to inform and improve conservation programs and wildlife management [6,16,27–29,58]. WERS data could be a valuable tool for local government areas and other land managers to pinpoint which

species are more impacted in their area and which threats are more prominent for these species, specifically for areas with high number of wildlife emergencies and high number of species involved [16]. This could help them to draft appropriate management plans, in line with other researchers' suggestions to use rehabilitation records to improve conservation efforts [58]. For example, flying foxes appear particularly vulnerable to human threats within the Greater Melbourne area, with entanglements as the most frequent threat. Similarly, Scheelings and Frith [5] identified unsafe fruit netting as a main cause of concern for flying foxes in Victoria and highlighted the need for increased public awareness of the dangers associated with such netting. Taylor-Brown et al. [6] have also called attention to the need for consistent, overarching policy to guide land management practices, including converting barbed-wire livestock fencing and monofilament netting into wildlife-friendly options. Other solutions have been put forward to reduce other threats; for example, reducing vehicle density and speed and increasing signage to reduce the impact of vehicle collisions [6] or controlling pets and a reduction in stray populations to ensure reduced the occurrence of pet attacks on wildlife [58]. WERS data could help identify where those solutions need to be investigated. Using WERS data to mitigate threats to wildlife will be more successful in combination with surveys aimed at understanding the public's values, believes and attitudes towards wildlife and wildlife management. This kind of surveys have allowed the identification of public support for management (e.g., pet containment or leashing) and need for education around the impact of human activities on wildlife in Australia and elsewhere [17,60–64].

WERS data could be used by practitioners not only to pinpoint the main species affected by human–wildlife interactions and the main threats to specific species, but also to monitor the impact of new interventions, policies and education campaigns aimed at mitigating these threats. For example, WERS records have been used in Australia to evaluate the success of a campaign designed to reduce threats impacting koalas [65]. The authors showed that the increased number of koala reports to the local WERS (RSCPA) was linked to better awareness of issues impacting koalas as a result of the campaign, and that the number of koalas reported as victims of car collisions and dog attacks to the WERS declined.

*4.2. WERS Demand and Response*

The number of cases reported to the WERS increased 2.5 fold over a ten-year period. This rise has been observed in other studies analysing WERS reports in Australia and internationally across various species, suggesting these patterns are most likely global [6,16,44]. This may result from increasing population and urbanisation, a growing awareness of WERS (e.g., social media and other outreach efforts), and/or a greater concern for animal welfare [6,16,55]. With the projected increases in population, an increase in wildlife reports and admissions is expected [6], further challenging WERS' capacity to respond to wildlife emergencies. The number of service providers is not increasing, which means that the number of animals and community members receiving support decreases. This creates an increased reliance on other service providers (e.g., veterinarians' pro bono services), which adds to their existing duties and might create more conflicts in the community. Indeed, the costs of assessing wildlife for veterinarians is significant; in an Australian state alone, such costs are evaluated at more than a million AUD annually [66]. Veterinarians are not always adequately trained to assess and temporarily care for wildlife, a problem that has been recognised in Australia [66] and elsewhere [67]. WERS data are able to shed light on which cases (species, cause types) they are more likely to encounter, which can assist in developing adequate training program for them.

Typically, WERS receive little or no on-going government funding [9] and heavily rely on volunteers to rescue and rehabilitate wildlife [37]. Barriers to volunteer retention and recruitment are numerous [9,36]. Stronger support and incentives for volunteers to assist WERS achieve their mission is needed in response to increasing demand [9]. Our analyses allowed for an understanding of which cases might be less likely to get a service

provider because of the associated species cause type or location, which might help to provide adequate training and communications to volunteers.

Service providers were found in slightly under 70% of cases requiring an intervention. Service providers were proportionally harder to find in Greater Melbourne, for some cause types and for some species. It could be harder to find service providers (the majority of which are rescuers) in Greater Melbourne compared to more rural settings due to an lack of interest or knowledge of wildlife in urban settings [68]. Carers could also be harder to find in the Greater Melbourne area due to the requirements of rescue centres in terms of space and facilities. An unfamiliarity with specific species' needs or lack of popularity of some species might decrease the likelihood of finding a service provider. Studying the reasoning behind the decision of a rescuer or carer to accept a case or not is needed to shed light on those findings and might highlight specific training needs for volunteers. Encouragingly, service providers were more likely to be found for cases involving threatened species and for the most frequent and/or deadly types of threats (e.g., wildlife hit by vehicle, entangled and attacked by cat).

Annually, more than 20,000 WERS cases require intervention in Victoria. Of the cases requiring intervention and having known outcomes, wildlife individuals in nearly one in five cases was euthanased. This shows that services providers have a prominent role in relieving the suffering of animals through euthanasia. In our dataset, nearly one-third of animals died or were euthanised. Other studies have highlighted a high rate of negative outcomes using WERS reports in Australia [41,69]. This result also highlights the high mortality linked with human threats to wildlife; this is a significant underestimation, however, due to the high percentage of cases without outcomes, but also due to thousands of cases being left unattended every year. This number would also increase if data from carers and rescuers who operate independently or in small local groups was captured. Given the fact that urbanisation worldwide negatively impacts wildlife, affecting billions of animals every year [10] and that WERS are involved in the rescue of millions of animals annually worldwide [20], these patterns are likely to be global.

In our dataset, mammals (kangaroos for example) tended to have poorer outcomes compared to birds overall and more animals were euthanased outside of Greater Melbourne. The worst outcomes were for animals hit by vehicles. Analyses of WERS records have shown that more extensive damage are linked to vehicles collisions [6,8], which occur more often on rural areas [70]. For animals suffering extensive damage, readaptation to life in the wild is more likely to be compromised and they are often euthanased [25], which happens more often outside Greater Melbourne in our dataset. Understanding which types of human–wildlife interactions result in the most negative outcomes and which species might fare better than others allows to prioritise funding allocation for threat mitigation.

Overall, one in five reports resulted in WERS providing education to the member of the public, indicating WERS substantial role in supporting the community to understand human-animal interactions. Themes that WERS operators regularly educate members of the public on include responsible pet ownership, the ecological importance of some species, the importance of not feeding wildlife, and the negative impacts of urbanisation on wildlife (Wildlife Victoria, personal communication). This complements the work of others showing the value of WERS for education of the community on wildlife issues [21,71]. Furthermore, education/advice given was an outcome that was more frequent in Greater Melbourne compared to the rest of the state. The high requirement for education provision in a highly urbanised environment perhaps highlights a lack of understanding people living in cities have in regard to wildlife [72]. A lack of understanding of the status or behaviour of wildlife encountered in urban environments might also indicate that education is needed specifically in cities. Urban residents often expect assistance to deal with wildlife interactions in cities, particularly when it comes to ill, injured, orphaned or 'nuisance' animals [23]. Our case study showed that educating the community in Greater Melbourne needs to focus on issues of responsible cat ownership whilst in the rest of the state, addressing vehicle collisions might be more important to mitigate human–wildlife interactions. For threatened species,

developing efficient messaging around wildlife-friendly netting and fencing (the main source of entanglements for flying foxes, the main threatened species in the dataset) will go a long way in making progress to conserve urban threatened species locally [5,6].

### 4.3. Limitations of This Study

Our dataset best speaks to the rescue rather rehabilitation services of WERS. For animals which need rehabilitation, often outcomes of the rehabilitation process are not linked back to the original Wildlife Victoria cases. Some WERS studies suggest that rehabilitation efforts might differ in success based on the characteristics of the animal or their length of stay in rehabilitation centres [24,73]. There are, however, important knowledge gaps due to the difficulty and costs of performing post-release monitoring [58] and a lack of systematic, large-scale, long-term and multi-species studies investigating the positive and negative impacts of wildlife rehabilitation. This hinders our understanding of how wildlife rehabilitation might be able to help mitigate human–wildlife interactions.

WERS operators are often under time constraints (responding collectively to hundreds of calls a day in busy periods), which can prevent them from obtaining detailed case information. These time constraints mean that operators sometimes make mistakes, or have limited time to follow up on cases or to seek more information on outcomes [7]. Responding to new calls and attending animals as quickly as possible take priority over data collection [7]. The information provided by WERS is not independently checked to reduce errors and a high proportion of cases do not have cause types as often members of the public do not directly witness the source of harm [11].

Outcomes were unknown for over half of cases where an intervention was required. Indeed, during busy times, rescuers, carers and veterinarians dealing with a high number of cases might not report back on the cases they have taken on. Even when the WERS knows that an animal has been released in the wild, it is not always clear whether it has been released in the same place as it was found. Although WERS recognise the importance of collecting data on the outcomes of the cases reported to them, they often lack the resources to follow up on cases. Another complication in collecting outcome information is that different service providers might use different case numbers for the same case (e.g., the vet vs. the carer) and separate systems to record their cases. A standardised system where a unique case number follows individual animals through the whole rehabilitation process is needed to make the best use of the data collected by different service providers for the same animals.

A reporting bias towards certain species might explain the over-representation of birds and mammals over other taxonomic groups; reporting biases towards more visible or diurnal, charismatic and non-threatening animals [6,55], or towards animals that are considered dangerous [14,71], have been suggested in other studies. Additionally, animals that are already dead might not be reported [69] (except for a few exceptions, e.g., marsupials in Australia, where a pouch check might be needed to ensure no joey is left behind). WERS data only capture some human and non-human threats to wildlife, as it focuses on some vertebrate species (in most cases, birds, mammals, to a lesser extent reptiles and amphibians, excluding fishes) and therefore it is best used in conjunction with data collected by ecologists using more traditional methods, and other citizen science programs focusing on different groups. Rather than these limitations being a reason to avoid using the data, it is greater encouragement for improved standards for data collection, especially given the vast amount of data that WERS collect on a daily basis.

We do acknowledge that with a rich database like ours, more complex, predictive analyses can be performed to provide insights into broad (long-term, state-wide) patterns of human–wildlife interactions. Rather than exploring predictive research questions, our goal was to give an overview of not only our dataset, but how WERS are able to respond to human and non-human threats to wildlife, and how this data could be used by others to help reduce threats to wildlife. Therefore, we argue that a simpler, more descriptive presentation of the results might be more appealing to local land managers (e.g., LGAs and

not-for-profits), which might focus on relevant subsets of the data to address specific issues (e.g., determining the main species impacted in their area and the main threats they face) without the need for complex statistical tools.

### 4.4. Implications of This Study and Recommendations

Following on from the discussion above, we discuss the implications of this study and highlight some barriers and opportunities to understand and mitigate human and non-human threats to wildlife:

1. Human–wildlife interactions affect a wide range of species and lead to the death of at least thousands of wild animals every year in a single state in Australia; they result in volunteers rescuing and rehabilitating thousands more, which comes at significant financial and mental health costs to the community who takes on this work.

2. WERS provide a valuable service to the community and wildlife [55], especially in urban environments, and are involved in the rescue of millions of animals throughout the globe annually [20]. Our case study showed that the main state-based WERS in Victoria is struggling to meet an increasing demand, a situation that may be similar in other parts of the world given the scale of population growth and urbanisation. In order for WERS to keep serving the community and wildlife in need of assistance, adequate resources need to be provided to them. With more resources, WERS would be able to better capture accurate data, including outcomes of reports.

3. WERS provide an invaluable opportunity to collect data to understand human and non-human threats to wildlife, which is increasingly recognised [6,11,14,27,54]. However, WERS data are underutilised. They should contribute to our understanding of the magnitude of human–wildlife interactions, of the pressure those threats put on WERS and the community and of the extent to which WERS can respond to them.

4. We have highlighted a need for data-driven education to mitigate specific threats that are locally relevant. Whilst WERS play a significant role of education on wildlife issues in the community worldwide [55,58,71], we caution that education should not be left for WERS alone to provide as they lack resources. Communication between WERS and other organisations (NGOs, schools, local government, etc.) who have some capacity to provide education will ensure that the most frequent and pressing issues are addressed community-wide.

**Supplementary Materials:** The following supporting information can be downloaded at: https://www.mdpi.com/article/10.3390/d15050683/s1, Table S1: Species common name, Latin name, native vs. non-native status and threatened status in accordance with the Flora and Fauna Guarantee Act (FFG) 1988 for the 573 or group of species reported (of which 443 were actual, identified species) for cases reported to Wildlife Victoria between 2010 and 2019. Table S2: Overview of the analysed dataset by Local Government Areas (LGAs) indicating the top reported non-threatened species, top cause type for top reported non-threatened species, top threatened species, top cause type for threatened species, total number of species reported, number of threatened species reported, whether the LGA is part of the Greater Melbourne area, and list of threatened species reported for each LGA of the dataset; Wildlife Victoria cases reported from 2010 to 2019. Figure S1: (A) Number of wildlife reports received by Wildlife Victoria from 2010 to 2019 (*n* = 333,796); (B) estimated Victorian resident population (Australian Bureau of Statistics data) from 2010 to 2019.

**Author Contributions:** Conceptualization, E.C.M.C. and A.P.A.C.; methodology, E.C.M.C. and A.P.A.C.; software, M.K and E.C.M.C.; validation, M.K. and E.C.M.C.; formal analysis, E.C.M.C.; investigation, E.C.M.C.; resources, E.C.M.C.; data curation, M.K.; writing—original draft preparation, E.C.M.C. and A.P.A.C.; writing—review and editing, E.C.M.C. and A.P.A.C.; visualization, M.K. and E.C.M.C.; supervision, A.P.A.C.; project administration, E.C.M.C. All authors have read and agreed to the published version of the manuscript.

**Funding:** This research received no external funding.

**Institutional Review Board Statement:** Not applicable.

**Data Availability Statement:** Data unavailable due to the data provider's restrictions. However, data can be visualised on the https://www.wildlifevictoria.org.au/wildlife-information/historical-data, accessed on 5 March 2023.

**Acknowledgments:** We wish to thank Megan Davidson, former CEO of Wildlife Victoria, for providing access to the data and David Wakeling, former Wildlife Victoria Operations Support Officer, for helping retrieve the data. In addition, we thank Wildlife Victoria's emergency response operators for collecting the data we have used in this paper. Alongside Wildlife Victoria's volunteers and other service providers, they work tirelessly, under stressful conditions, to provide excellent customer service, support and educate the community and assess, rescue and rehabilitate animals in need. We also thank the members of the public who call the Wildlife Victoria and similar services for their willingness to help wildlife and report sick, orphaned and injured animals. We also wish to thank the three reviewers who took the time to provide detailed comments and suggestions to improve the manuscript.

**Conflicts of Interest:** The authors declare no conflict of interest.

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
