# Peer review of "Wildlife Emergency Response Services Data Provide Insights into Human and Non-Human Threats to Wildlife and the Response to Those Threats"

_diversity, doi:10.3390/d15050683_

Round 1

Reviewer 1 Report

The authors address an important topic using an appropriate data set. The following comments may assist in revising the paper. 

1.  The authors take a common line in describing some interactions between humans and fauna as conflict. Peterson et al. Pacific Conservation Biology 19, 94-103 argue convincingly that characterising these interactions as conflict casts wildlife as conscious human antagonists, which is likely untrue. Whether or not one accepts this view, it is worth acknowledging before accepting the conflict paradigm, with its associated assumptions.

2. As stated in the paper at the moment, human-wildlife conflict is very broadly defined. Why is an attack by a cat an example of human-wildlife conflict? Could the cat be a stray and not a pet?

3. It was unclear to me how the expected values for the chi-squared tests were determined. This should be explained. Also, what software was used?

4. Line 178: chi-square goodness of fit tests. 

5. Line 181: significance at the 5% level. or alternatively p < 0.05, not p = 0.05.

6. Do you wish to speculate on the management implications of the findings? For example, does the prevalence  of cat attacks support cat management policies such as containment? Or are these unlikely to be helpful, given that dogs (which must be contained), are also significant causes of wildlife presentation?

Author Response

The authors address an important topic using an appropriate data set. The following comments may assist in revising the paper. 

  1.  The authors take a common line in describing some interactions between humans and fauna as conflict. Peterson et al. Pacific Conservation Biology 19, 94-103 argue convincingly that characterising these interactions as conflict casts wildlife as conscious human antagonists, which is likely untrue. Whether or not one accepts this view, it is worth acknowledging before accepting the conflict paradigm, with its associated assumptions.
  • We agree with this point and have replaced framing of conflict with ‘human and non-human threats to wildlife’ when looking at threats that can be anthropogenic (e.g. collision with vehicles) or not (e.g. severe weather, interactions with other animals) [throughout]. At times, we also more specifically refer to ‘human-wildlife interactions’ as suggested in Peterson et al. 2013 [throughout] and we include that reference [line 70 of the manuscript with tracked changes]. We hope this clarifies the framing of the types of threats and interactions we present in the paper.
  1. As stated in the paper at the moment, human-wildlife conflict is very broadly defined. Why is an attack by a cat an example of human-wildlife conflict? Could the cat be a stray and not a pet?
  • Thank you for pointing this out. The terms ‘human-wildlife conflicts’ have now been replaced as per our response to point number 1. Moreoever, we now specify that the threats were are invesitgating can be directly caused by humans and their activities, or could be due to other factors (e.g. severe weather, interactions with other animals) [line 115-116 & 517-518 of the manuscript with tracked changes].
  1. It was unclear to me how the expected values for the chi-squared tests were determined. This should be explained. Also, what software was used?
  • We now provide a short explanation about the expected values in the chi-square test. We include reference to R software. [lines 226-238 of the manuscript with tracked changes]
  1. Line 178: chi-square goodness of fit tests. 
  • Thanks, this is changed. [line 226 of the manuscript with tracked changes]
  1. Line 181: significance at the 5% level. or alternatively p < 0.05, not p = 0.05.
  • Corrected [line 238 of the manuscript with tracked changes]
  1. Do you wish to speculate on the management implications of the findings? For example, does the prevalence  of cat attacks support cat management policies such as containment? Or are these unlikely to be helpful, given that dogs (which must be contained), are also significant causes of wildlife presentation?
  • This is a good point, thank you. We have now added some information in the discussion to discuss the management implications of the findings. We think that local land managers might benefit from using WERS data to identify the main species impacted, and how those are impacted, and we present potential solutions (e.g. wildlife-friendly fencing and netting for flying foxes, reduced speed/increase signage for car collisions, education such as the koala campaign to reduce attacks by dogs and car collisions, etc). We hope that these examples significantly enrich the discussion. [line 571-579 of the manuscript with tracked changes]

Reviewer 2 Report

Review of ”Wildlife Emergency Response Services data provide insights into human-wildlife conflicts and their mitigation

The authors have provided a comprehensive analysis of data from a WERS in Australia, extracting relevant information to comment on the effects on wildlife in the local area. 

This is a relevant addition to our current knowledge on wildlife and conservation efforts through wildlife rehabilitation.  

I have a few general comments: 

-        There is a substantial need for subtitles throughout the manuscript.

-        The layout should be adapted to the MDPI submission format and placed in the template

-        Are we really dealing with human-wildlife conflicts as such by looking into rehabilitation data? Are we talking conflicts when the causes for admission are e.g. orphaned wildlife? Or fledglings? And is it really a human-wildlife conflict per definition if a flying dog is entangled in netting? In my professional network human-wildlife conflicts are defined as e.g. lions attacking livestock or human family members, and therefore, they are hunted by humans. 

-        It appears you identified 21 causes for calls (Table 1) including extreme weather and orphans. Did you exclude these from your analyses? Because that would not be human-wildlife conflict in my view? Or did you simply analyse all 21 causes? Correct me if I am wrong, but I could not find any description of this in the Method section. 

-        The authors state several times that this is the first study to investigate how these human-wildlife conflicts can be mitigated through wildlife rehabilitation. This is not at all the case? There are several publications similar to this one describing wildlife rehabilitation and the outcomes and causes for admission. Even though I think the study is relevant it is by no means groundbreaking or innovative even though the authors appear to try and make it seem so. And to be honest, I am not sure the article even really replies to the question about how human-wildlife conflicts can be mitigated through wildlife rehabilitation? Other than describing what causes are most often responded to and handled. Perhaps it would be relevant to instead simply state that you describe the cases of wildlife rehabilitation in the area and try to extract relevant information out of this instead of pretending to try and solve a problem, which you do not do with this particular manuscript? 

-        I lack solutions based on the findings of the analyses. I consider the suggestions provided and the discussion of the results somewhat superficial and not really contributing with much relevant knowledge or initiatives. 

-        I would like to see a discussion of the potential impact of wildlife rehabilitation on the local wildlife species (both positive and negative), especially given the many cases handled per year.

-        You do not seem to discuss any potentially negative effects of wildlife rehabilitation such as long-term stress, exposure to disease from other individuals in care etc. which I think would also be relevant.

-        In relation to that, it is also worth discussing whether a standardization/legislation on the practice of wildlife rehabilitation would be relevant to improve the quality of care provided and hence the number of positive outcomes. 

-        You refer to a Table A1, would that be Table S1 (Supplementary Material)? Then please edit it to fit the MPDI format. It just seems a bit sloppy that the manuscript is not adjusted to meet the formatting of MDPI (e.g. placed in the template) before submission. 

Specific comments: 

L. 54: more? That is a bit unspecific. I would suggest adding incidents in relation to gardening which are also very common, e.g. Rasmussen, S.L.; Schrøder, A.E.; Mathiesen, R.; Nielsen, J.L.; Pertoldi, C.; Macdonald, D.W. Wildlife Conservation at a Garden Level: The Effect of Robotic Lawn Mowers on European Hedgehogs (Erinaceus europaeus). Animals 2021, 11, 1191. https://doi.org/10.3390/ani11051191

L. 55-57: How do you plan to solve this through wildlife rehabilitation? They are not prevented but only handled (“damage control” you could say). 

L. 57: There are already several papers out there describing wildlife rehabilitation, causes of admission, outcomes, etc. so I do not think this sentence is appropriate. 

L. 58: Zoonoses go both ways. Wildlife can also be infected by humans. 

L. 62: Could you perhaps elaborate on the effect of this? Why would you say it was a problem? 

L. 63: Please provide references for this. There are studies describing the effects on wildlife of captivity during rehabilitation, e.g. increased stress levels in hedgehogs in care compared to wild hedgehogs (this paper also cites several papers about effects of handling/captivity in wildlife): Rasmussen, S.L., Kalliokoski, O., Dabelsteen, T. et al. An exploratory investigation of glucocorticoids, personality and survival rates in wild and rehabilitated hedgehogs (Erinaceus europaeus) in Denmark. BMC Ecol Evo 21, 96 (2021). https://doi.org/10.1186/s12862-021-01816-7

L. 64: Please define WERS

L. 82: Please explain this.

L. 87: I am not a native English speaker, so please forgive me if I am mistaken, but shouldn’t it be “records”? 

L. 147-49: I am not sure I understand the sentence. Could you please consider rephrasing this? 

L. 160: Is the WERS you analysed data from, Wildlife Victoria, open around the clock? And are the phones handled by volunteers or employed professionals? 

L. 195: Please introduce what FFG is

L. 195 onwards: The sentence is 5 lines long! What is meant by “reported in the whole database”? Could you please rephrase/explain and shorten the sentence. 

L. 203: Didn’t you exclude invertebrates or did I misunderstand something?

L. 249: Please define impacted? Does it mean fewer cases with this species? 

L. 272: Could you please add the information about % here? 

L. 277: Were they released on the spot? 

L. 281: This is interesting. Please discuss why they were euthanised (in the discussion session). Why do you think there were more of these cases in the rest of the state compared to Melbourne? 

L. 283: Please add %. 

L. 290: Compared to what?

L. 291: Compared to what? 

L. 307: Is this really the only cause? How about awareness, influence of SoMe outreach/visibility during the years? When did the WERS start? (perhaps this should be in the discussion, but I do think it is relevant to include)

L. 312: Attached to? That is strange wording if you ask me. It continues throughout the section. I would suggest a change of word. 

L. 349: You write “such as”. Are there more? Then they should be listed.

L. 358: E.g.- again, are there more, then please list them.

L. 397: Why “common” and not just common?

L. 398: Please rephrase this, something seems odd grammatically. I at least have challenges understanding this correctly. 

L. 414: What were the others? Are they listed somewhere, then please refer to this list. Table 1 should be the right one, right? 

L. 423: Problematic? Do you mean lethal or frequent here? 

L. 425: What do you mean by “fixed infrastructure”? 

L. 460: What about awareness of the rescue centre (when did it start, did SoMe influence the awareness as seen with so many other rescue centres around the world)? And a general shift in people’s fondness and attention to wildlife during more recent years? 

L. 475: Why is this? Could it be explained by the requirements for a wildlife rescue centre in terms of space, facilities etc. which makes it more likely to be situated in a less urban setting? (It is often farms or the like housing rescue centres) 

L. 488: I would refer to wildlife individuals or something similar instead of cases. 

L. 500: Perhaps this is due to more extensive damage cause by vehicle collisions compared to netting injuries? I think you should discuss this somehow instead of just more or less listing the results. 

L. 521: Previous research has described how vehicle collisions with wildlife more often happen in rural areas or less trafficked areas, which may explain these findings. I think you should make a reference to this and discuss this. One example: Rasmussen, S.L.; Berg, T.B.; Martens, H.J.; Jones, O.R. Anyone Can Get Old—All You Have to Do Is Live Long Enough: Understanding Mortality and Life Expectancy in European Hedgehogs (Erinaceus europaeus). Animals 2023, 13, 626. https://doi.org/10.3390/ani13040626

L. 550: You mention “some vertebrate species”. Which ones were left out then? 

In the discussion I would like to read about the percentage success rate of outcomes (how many individuals were successfully treated and released back into the wild), which I believe would constitute your definition of human-wildlife conflict mitigation= the whole point of the manuscript, right? And then followed by a discussion about this success rate, please. 

Author Response

Review of ”Wildlife Emergency Response Services data provide insights into human-wildlife conflicts and their mitigation”

The authors have provided a comprehensive analysis of data from a WERS in Australia, extracting relevant information to comment on the effects on wildlife in the local area. 

This is a relevant addition to our current knowledge on wildlife and conservation efforts through wildlife rehabilitation.  

I have a few general comments: 

-        There is a substantial need for subtitles throughout the manuscript.

  • Subtitles added [line 51, 78, 146, 155, 217 of the manuscript with tracked changes]

-        The layout should be adapted to the MDPI submission format and placed in the template.

  • We have adapted the layout to the MDPI format.

-        Are we really dealing with human-wildlife conflicts as such by looking into rehabilitation data? Are we talking conflicts when the causes for admission are e.g. orphaned wildlife? Or fledglings? And is it really a human-wildlife conflict per definition if a flying dog is entangled in netting? In my professional network human-wildlife conflicts are defined as e.g. lions attacking livestock or human family members, and therefore, they are hunted by humans. 

  • As per your and other reviewers’ suggestions, the framing of the paper has been changed. We have replaced framing of conflict with ‘human and non-human threats to wildlife’ when looking at threats that can be anthropogenic (e.g. collision with vehicles) or not (e.g. severe weather, interactions with other animals) [throughout]. At times, we also more specifically refer to ‘human-wildlife interactions’ as suggested in Peterson et al. 2013 [throughout]. We hope this clarifies the framing of the types of threats and interactions we present in the paper.

-        It appears you identified 21 causes for calls (Table 1) including extreme weather and orphans. Did you exclude these from your analyses? Because that would not be human-wildlife conflict in my view? Or did you simply analyse all 21 causes? Correct me if I am wrong, but I could not find any description of this in the Method section. 

  • Thank you for pointing this out. You are right, though most threats to wildlife that we identified were anthropogenic (top 10 causes), other causes were included (analysed) which might not fit the definition of ‘human-wildlife conflicts’ (e.g. severe weather, interactions with other animals). As discussed above, we have now changed the framing of the paper to discuss the value of WERS data for identifying ‘human and non-human threats to wildlife’ [throughout]. We also make it clear that the majority of cases are caused by human-wildlife interactions but acknowledge that some are not [line 115-116 & 517-518 of the manuscript with tracked changes]. In the methods we now clarified which of the 21 causes were analysed (i.e. the top 10) [line 231-232 of the manuscript with tracked changes].

-        The authors state several times that this is the first study to investigate how these human-wildlife conflicts can be mitigated through wildlife rehabilitation. This is not at all the case? There are several publications similar to this one describing wildlife rehabilitation and the outcomes and causes for admission. Even though I think the study is relevant it is by no means groundbreaking or innovative even though the authors appear to try and make it seem so.

  • Our intention was not to claim that this is the first study to investigate how human-wildlife conflicts can be mitigated through wildlife rehabilitiation. Rather we are suggesting that few studies have shown the capacity for WERS data to be used to understand broad multi-species and multi-threat issues. This study calls for a more systematic use of such data and its integration into wildlife management processes. We recognise and reference several articles which utilise WERS data to investigate species specific threats or threat specific issues. Since submitting the original manuscript we’ve identified some new publications that contribute to this discussion. There are still only a few studies which investigate broad patterns across species and threats (e.g. Vezyrakis 2023, Pop et al 2023, Long et al 2020, Taylor brown et al 2019), though most of those do not use emergency hotline data, but rather wildlife rehabilitation data. We now referenced these in the latest version of the manuscript. In addition, we were using phrasing like “reactive measures, e.g. rescue and rehabilitation””; to clarify, we have now removed “rehabilitation” to make it clear that our dataset mostly focuses on cases for rescues (only about a fifth of cases require rehabilitation) [line 30 of the manuscript with tracked changes].We also state more clearly that we are looking at how the WERS respond, rather than mitigate conflicts [title, line 64-66, 733, 761]. We could not find any studies specifically investigating how WERS are able to respond to the cases reported to them, but we are happy to include them if you have any suggestions of studies that are looking at this.

And to be honest, I am not sure the article even really replies to the question about how human-wildlife conflicts can be mitigated through wildlife rehabilitation? Other than describing what causes are most often responded to and handled. Perhaps it would be relevant to instead simply state that you describe the cases of wildlife rehabilitation in the area and try to extract relevant information out of this instead of pretending to try and solve a problem, which you do not do with this particular manuscript? 

  • As mentioned above, we have changed some of the framing, and have removed the part about mitigation in the title. Our focus, however, is not on wildlife rehabilitation, but more so, wildlife rescue. We focus on how our local WERS is able to respond to threats to wildlife (what we called included in the parts talking about “mitigation” before) [see point above]

-        I lack solutions based on the findings of the analyses. I consider the suggestions provided and the discussion of the results somewhat superficial and not really contributing with much relevant knowledge or initiatives. 

  • Thank you. In line with this comment and that of other reviewers, we are discussing the results in the context of existing literature more, and included more information about solutions (including how WERS data could be used to mitigate human-wildlife interactions and monitor the success of new policies, actions and campaigns) [line 551-579 of the manuscript with tracked changes]. We hope this significantly enirches the discussion. When it comes to how the WERS is able to respond to widllife emergencies, we had already presented solutions (increased funding, increased training for volunteers, shift education duties to other organisations rather than WERS which have limited capacity) [line 615-637, 765-768].

-        I would like to see a discussion of the potential impact of wildlife rehabilitation on the local wildlife species (both positive and negative), especially given the many cases handled per year.

  • This research discusses WERS data (emergency call data) with most cases never requiring wildlife rehabilitation intervention. While we absolutely agree a deeper discussion of wildlife rehabilitation is valuable, it is beyond the scope of this research, which is focused specifically on what insights we can gain about human-wildlife interactions from emergency call data. There are also important knowledge gaps regarding both positive and negative outcomes of rehabilitation on local widllife and typically, studies that have investigated this in our region (and elsewhere, as far as we know) have had small sample sizes and/or focused on one specific threats and/or more often on a specific species. This hinders discussion of broad patterns (what we focus on) in relation to wildlife rehabilitation. We have now clarified in the methods exactly which part of the human-animal interaction WERS data best speaks to [line 171-174 of the manuscript with tracked changes] and talked about the points above in the discussion [line 690-698 of the manuscript with tracked changes].

-        You do not seem to discuss any potentially negative effects of wildlife rehabilitation such as long-term stress, exposure to disease from other individuals in care etc. which I think would also be relevant.

  • As above, a discussion of wildlife rehabilitation is beyond the scope of this research and there are significant knowledge gaps when it comes to understanding the effects of wildlife rehabilitation.

-        In relation to that, it is also worth discussing whether a standardization/legislation on the practice of wildlife rehabilitation would be relevant to improve the quality of care provided and hence the number of positive outcomes. 

  • As above, a discussion of wildlife rehabilitation is beyond the scope of this research and there are significant knowledge gaps when it comes to understanding the effects of wildlife rehabilitation.

-        You refer to a Table A1, would that be Table S1 (Supplementary Material)? Then please edit it to fit the MPDI format. It just seems a bit sloppy that the manuscript is not adjusted to meet the formatting of MDPI (e.g. placed in the template) before submission. 

  • Thank you for pointing this out. Now edited to the MPDI format [line 183, 251-252, 257, 302, 399, 817 and below of the manuscript with tracked changes].

Specific comments: 

  1. 54: more? That is a bit unspecific. I would suggest adding incidents in relation to gardening which are also very common, e.g. Rasmussen, S.L.; Schrøder, A.E.; Mathiesen, R.; Nielsen, J.L.; Pertoldi, C.; Macdonald, D.W. Wildlife Conservation at a Garden Level: The Effect of Robotic Lawn Mowers on European Hedgehogs (Erinaceus europaeus). Animals202111, 1191. https://doi.org/10.3390/ani11051191
  • This sentence is now changed and this point is now acknowledged, with the reference mentioned added [line 60-62 of the manuscript with tracked changes].
  1. 55-57: How do you plan to solve this through wildlife rehabilitation? They are not prevented but only handled (“damage control” you could say). 
  • We agree that wildlife rehabilitation is indeed “damage control”. Our intention was not to claim that these problems can be “solved” by wildlife rehabilitation (though as mentioned, we now put a bigger emphasis on rescue rather than rehabilitation to avoid confusion). What we mean here is that threats might be reduced (e.g. an animal trapped in a building can be rescued and released back to its natural habitat through the rescue process) if and when WERS are able to respond to cases threatening wildlife. We have rephrased this sentence with a more nuanced language and an emphasis on both characterising the threats and responding to the threats [line 64-66 of the manuscript with tracked changes].
  1. 57: There are already several papers out there describing wildlife rehabilitation, causes of admission, outcomes, etc. so I do not think this sentence is appropriate. 
  • The phrasing of this sentence has now been changed as mentioned above. We did not intend to claim that we are the first paper describing wildlife rehabilitation, causes of admission or outcomes. Rather, we mean that as far as we know, there are not a lot of studies (corrected from “no studies” to take into account recent studies) that take a systematic, broad (multi-location, multi-taxomic) approach to look at how wildlife rescue service data can help understand human and non-human threats to wildlife. There also are very few that look at the extent of which wildlife response services can respond to emergencies [line 64-66 of the manuscript with tracked changes and elsewhere, as metnioned above].
  1. 58: Zoonoses go both ways. Wildlife can also be infected by humans. 
  • This is now acknowledged. [line 75 of the manuscript with tracked changes]
  1. 62: Could you perhaps elaborate on the effect of this? Why would you say it was a problem? 
  • We do not imply that this is a problem (we have now removed “less recognised” from this sentence to clarify). Rather, here, we provide context and describe the type of negative human-willdife interactions, both from the human and wildlie perspective [line 73-74 of the manuscript with tracked changes].
  1. 63: Please provide references for this. There are studies describing the effects on wildlife of captivity during rehabilitation, e.g. increased stress levels in hedgehogs in care compared to wild hedgehogs (this paper also cites several papers about effects of handling/captivity in wildlife): Rasmussen, S.L., Kalliokoski, O., Dabelsteen, T. et al.An exploratory investigation of glucocorticoids, personality and survival rates in wild and rehabilitated hedgehogs (Erinaceus europaeus) in Denmark. BMC Ecol Evo21, 96 (2021). https://doi.org/10.1186/s12862-021-01816-7

  • At this point in the text, we have not introduced wildlife rescue or rehabilitation. We are talking about the kind of human-wildlife interactions which can take place, therefore, we do not think this reference is very fitting here. However, we have included it when we briefly discuss wildlife rehabilitation (not our main focus, as stated above, therefore we are not discussing at length the impacts of captivity on wildlife; we do mention in the discussion) [line 690-698 of the manuscript with tracked changes]. We have added other references here, however, as we agree this point is important and needs to be supported by references [line 75-76 of the manuscript with tracked changes].
  1. 64: Please define WERS
  • This is now defined [line 80 of the manuscript with tracked changes].
  1. 82: Please explain this.
  • To clarify our point, two sentences now provide examples of education provided by WERS operators [line 101-105 of the manuscript with tracked changes].
  1. 87: I am not a native English speaker, so please forgive me if I am mistaken, but shouldn’t it be “records”? 
  • We have now changed this to “records” [line 75-76 of the manuscript with tracked changes].
  1. 147-49: I am not sure I understand the sentence. Could you please consider rephrasing this?
  • This information has now been rephrased. We hope this clarify which kind of data was used for which analysis [line 110 of the manuscript with tracked changes].
  1. 160: Is the WERS you analysed data from, Wildlife Victoria, open around the clock? And are the phones handled by volunteers or employed professionals? 
  • This is now clarified [line 158-163 of the manuscript with tracked changes].
  1. 195: Please introduce what FFG is
  • This is now defined here. [line 253-254 of the manuscript with tracked changes]
  1. 195 onwards: The sentence is 5 lines long! What is meant by “reported in the whole database”? Could you please rephrase/explain and shorten the sentence. 
  • This part is now broken down in two sentences and what we meant by “reported in the whole database is clarified” is clarified (both found within and outside of the Greater Melbourne area). [line 254-257 of the manuscript with tracked changes]
  1. 203: Didn’t you exclude invertebrates or did I misunderstand something?
  • We did for the statistical analyses (as now clarified in the text, those analyses focus only on the top 10 species) [line 231-232 of the manuscript with tracked changes], but we included them at first when we give an overview of the whole dataset. To clarify, the subheading for this section has been changed to “overview of species affected” [line 250 of the manuscript with tracked changes].
  1. 249: Please define impacted? Does it mean fewer cases with this species? 
  • This is now clarified. What we mean is that the number of cases logged is different. [line 309-310 of the manuscript with tracked changes]
  1. 272: Could you please add the information about % here? 
  • We are sorry, but we don’t understand this question as % as indicated.
  1. 277: Were they released on the spot? 
  • Unfortunately, we do not have that information. Though it is advised that animals are released where they were found, there are situations in which this is not practical or feasible (e.g. the habitat of this animal has been destroyed). This information does sometimes appear in the case notes (the analysis of which was beyond the scope of this study), but this information is not always recorded because of lack of time from volunteers and/or operators (as now mentioned in the discussion). [line 709-710 of the manuscript with tracked changes].
  1. 281: This is interesting. Please discuss why they were euthanised (in the discussion session). Why do you think there were more of these cases in the rest of the state compared to Melbourne? 
  • This is now acknowledged [line 660-662 of the manuscript with tracked changes].

  1. 283: Please add %. 
  • Whilst this section has been deleted (we no longer group cases by positive or negative outcomes, we now indicate the number of cases and percentages for the different (known) outcomes [line 368-371 of the manuscript with tracked changes].
  1. 290: Compared to what?
  • This section has now been deleted.
  1. 291: Compared to what? 
  • This section has now been deleted.
  1. 307: Is this really the only cause? How about awareness, influence of SoMe outreach/visibility during the years? When did the WERS start? (perhaps this should be in the discussion, but I do think it is relevant to include)
  • We did not intend to imply this was the only cause (partly the reason why this information is presented in supplementary material rather than the main text), but this is the only factor we have data for (so the only thing we can include in the result section; we do not have data for growing awarness linked to social media or other outreach efforts). However, we did acknowledge some of your points in the discussion. We have now added a bit more details to reflect your comments [line 593-595 of the manuscript with tracked changes]. We now indicate in the methods when Wildlife Victoria was created [line 147 of the manuscript with tracked changes].
  1. 312: Attached to? That is strange wording if you ask me. It continues throughout the section. I would suggest a change of word. 
  • Thank you for pointing this out. This was the language used by the WERS hence why we were using it, but we have now replaced this term by “linked to” [throughout].

  1. 349: You write “such as”. Are there more? Then they should be listed.
  • Yes, technically, there are more. However, what we focus on here (to avoid distracting the reader and make the take home messages less clear), are the main differences (i.e. the differences between the groups which have the HIGHEST differences from expectations). We now state this in the methods to calrify that we are not looking at the differences between every single group. [line 236-237 of the manuscript with tracked changes]
  1. 358: E.g.- again, are there more, then please list them.
  • See our response to the comment above re L349.
  1. 397: Why “common” and not just common?
  • Because some species can be THOUGHT to be common despite lack of data. To avoid confusion, we have removed the “” [line 495 of the manuscript with tracked changes].
  1. 398: Please rephrase this, something seems odd grammatically. I at least have challenges understanding this correctly. 
  • Thank you. This is now rephrased [line 497-498 of the manuscript with tracked changes].
  1. 414: What were the others? Are they listed somewhere, then please refer to this list. Table 1 should be the right one, right? 
  • This refers specifically to the MAIN causes (i.e. top 10) – this is now clarified [line 514 of the manuscript with tracked changes].
  1. 423: Problematic? Do you mean lethal or frequent here? 
  • We meant more frequent. This part has been deleted however.
  1. 425: What do you mean by “fixed infrastructure”? 
  • We mean fixed man-made structure such as a buildings, windows, poles or power lines. We have, however, deleted this part now as reviewers sugegsted shortening the discussion and/or being more conscious of not repeating results too much.
  1. 460: What about awareness of the rescue centre (when did it start, did SoMe influence the awareness as seen with so many other rescue centres around the world)? And a general shift in people’s fondness and attention to wildlife during more recent years? 
  • We agree that these factors might influence the growing number of cases. This is acknowledged at the start of this section (“This rise may result from increasing population and urbanisation, a growing awareness of WERS (e.g. social media and other outreach efforts), and/or a greater concern for animal welfare.”) [line 593-595 of the manuscript with tracked changes]; however, lower down, the references we include specifically consider population increase. To avoid making the discussion too lengthy, we do not repeat this information here.
  1. 475: Why is this? Could it be explained by the requirements for a wildlife rescue centre in terms of space, facilities etc. which makes it more likely to be situated in a less urban setting? (It is often farms or the like housing rescue centres) 
  • Thank you. This is a good point, which we now discuss. The majority of service providers are rescuers (and we now discuss why they might be harder to find in Melbourne), though we do acknwoledge your point regarding rescue centres [line 626-629 of the manuscript with tracked changes]
  1. 488: I would refer to wildlife individuals or something similar instead of cases.
  • This has now been changed [line 642 of the manuscript with tracked changes].  
  1. 500: Perhaps this is due to more extensive damage cause by vehicle collisions compared to netting injuries? I think you should discuss this somehow instead of just more or less listing the results. 
  • This is now acknowledged [line 656-663 of the manuscript with tracked changes].
  1. 521: Previous research has described how vehicle collisions with wildlife more often happen in rural areas or less trafficked areas, which may explain these findings. I think you should make a reference to this and discuss this. One example: Rasmussen, S.L.; Berg, T.B.; Martens, H.J.; Jones, O.R. Anyone Can Get Old—All You Have to Do Is Live Long Enough: Understanding Mortality and Life Expectancy in European Hedgehogs (Erinaceus europaeus). Animals202313, 626. https://doi.org/10.3390/ani13040626
  • This is now acknowledged and the reference mentioned has been included [line 659-663 of the manuscript with tracked changes].
  1. 550: You mention “some vertebrate species”. Which ones were left out then? 
  • Fish are not reported to WERS. This is now mentioned [line 724-725 of the manuscript with tracked changes]..

In the discussion I would like to read about the percentage success rate of outcomes (how many individuals were successfully treated and released back into the wild), which I believe would constitute your definition of human-wildlife conflict mitigation= the whole point of the manuscript, right? And then followed by a discussion about this success rate, please. 

  • Whilst we agree that this is an important point, we unfortunalety do not have access to this data. The rehabilitation process is usually separate to the WERS itself, and in most cases in our WERS database, the outcomes is unknown (even when the animal has been taken to a rehabilitation facility – which keeps their own records). This is acknowledged and we now reinforce the point that lack of data hinders a broader discussion of rehabilitation specifically [line 690-698 of the manuscript with tracked changes and points above].

Reviewer 3 Report

This is an interesting and well written paper examining a database from the Wildlife Emergency Response Services database from Australia. This is a unique chance for the authors to show a variety of wildlife conflicts reported in a systematic way. The work is incredibly descriptive, which is a shame since the sample size is huge and would allow for very nice statistical analyses. This weakens the paper but the data and topic remain interesting.

Line 58: I appreciate the definition here of human wildlife conflict. A number of people in this area of study are now avoiding the term conflict and instead use the term interactions. I disagree, as the many conflicts are just one range of interactions. To satisfy those people, however, you might want two lines on the types of HWI then can give your definition of conflict as a type of this interaction.

Line 95 - although it is introduced in the abstract WERS needs to be spelled out and introduced here in the text - I might start with the paragraph on line 85 so we know we are in Australia - perhaps also here something specific about Victoria (I know you go into it in the methods but just two lines about why it is an ideal case study) so we have the context - then the WERS paragraphs can go together

Line 111 and throughout - the term data is plural; spell out numbers ten or less

Line 110 - research questions - it would have been nice to use some of the literature review in the introduction to come up with less descriptive and more predictive research questions (or questions in the context of the literature) so we know why each of these important and even how that data to examine these issues are often lacking, showing a greater importance to the manuscript.

Figure 1 - the heading should stand alone from the text so again we should know where these data were collected too

Line 120 - I guess this should be 6.7 million?

Paragraph 202 - in my experience this journal has common names of species lower case

Line 249 - For me a chi square test is not appropriate for such a huge dataset. It would be much more interesting to use generalised linear mixed models where you could combine temporal data, like year; area of the conflict; taxonomic group etc. Here we could see interactions between the variables rather than a long list of descriptive stats. It is also not typical to report the observed and expected values. For degrees of freedom of one, you also need to do post hoc tests.

Line 283 - be wary with release to the wild being a positive outcome; this depends if all guidelines were followed, including health checks and post release monitoring for an extended period - release on its own and moving animals from say an urban setting where they know how to live their lives, to a rural setting, where they die almost immediately, would certainly not be positive.

The discussion is largely a repetition of the results and you have much more opportunity to discuss the work in the context of the literature - for example line 418 - what are these studies? How were the results consistent? Why? Line 429 - what studies can you recommend to help them with this? Has it ever worked? Line 443 - again have other urban residents in other places benefited? What evidence is there for this statement? Line 459 - again here you reference some studies but do not discuss your results in line with them -instead just reiterating your results.

Overall the discussion can be shortened in that the results do not need to be repeated but throughout need to be discussed in the context of the literature to help the practitioners see what changes are possible

Author Response

This is an interesting and well written paper examining a database from the Wildlife Emergency Response Services database from Australia. This is a unique chance for the authors to show a variety of wildlife conflicts reported in a systematic way. The work is incredibly descriptive, which is a shame since the sample size is huge and would allow for very nice statistical analyses. This weakens the paper but the data and topic remain interesting.

  • We discuss this point below, and whilst we have considered more complex statistical analyses, we have opted against them. This is because we think that managers for which this type of data is most relevant - e.g. local city councils, NGOs - would also use a descriptive approach to fit their needs (e.g. reducing a specific threat like vehicle collisions, or reducing threats for a specific species). We think that complex statistics are not accessible to these types of land managers, and using them could discourage them from accessing and using this kind of data. Rather than fully analysing patterns in our dataset, we focus on showing the potential of WERS data to understand human and non-human threats to wildlife, and to understand to what extent WERS is able to respond to those threats. This is now acknowledged [line 730-738 of the manuscript with tracked changes].

Line 58: I appreciate the definition here of human wildlife conflict. A number of people in this area of study are now avoiding the term conflict and instead use the term interactions. I disagree, as the many conflicts are just one range of interactions. To satisfy those people, however, you might want two lines on the types of HWI then can give your definition of conflict as a type of this interaction.

  • Thank you for this comment. Based on your and other reviewer comments, the framing of the paper has been changed. We have replaced framing of conflict with ‘human and non-human threats to wildlife’ when looking at threats that can be anthropogenic (e.g. collision with vehicles) or not (e.g. severe weather, interactions with other animals). At times, we also more specifically refer to ‘human-wildlife interactions’. We hope this clarifies the framing of the types of threats and interactions we present in the paper.

Line 95 - although it is introduced in the abstract WERS needs to be spelled out and introduced here in the text - I might start with the paragraph on line 85 so we know we are in Australia - perhaps also here something specific about Victoria (I know you go into it in the methods but just two lines about why it is an ideal case study) so we have the context - then the WERS paragraphs can go together

  • This is a good suggestion. We now spell out WERS, when it is first introduced [line 80 of the manuscript with tracked changes]. The paragraphs have now been joined into one and we indicate why Victoria is an ideal case study [line 122-124 of the manuscript with tracked changes].

Line 111 and throughout - the term data is plural; spell out numbers ten or less

  • Fixed throughout. Thank you.

Line 110 - research questions - it would have been nice to use some of the literature review in the introduction to come up with less descriptive and more predictive research questions (or questions in the context of the literature) so we know why each of these important and even how that data to examine these issues are often lacking, showing a greater importance to the manuscript.

  • We very much appreciate this point. We recognise that predictive research questions would provide valuable insights into human-wildlife interactions and the threats faced by wildlife. There are some papers that use species specific WERS data for predictive analysis, and more recently some that cover multiple species. Our intention with this manuscript was to highlight the potential of WERS data for understanding local and species specific threats. We believe that for most Local Government Organisations or NGOs (who could immediately benefit from WERS data) predictive analysis is not necessary, and may even be a hinderance, for gaining insights from these data. Much simpler descriptive methods could help the organisations to plan and manage human-wildlife interactions and threats to local wildlife. We believe that this point isn‘t well represented in the literature but is worth recognising. That was one of the main intentions of this research. We do hope to carry out further research and specifically answer predictive questions on the evolution of human and non-human threats to wildlife in other papers in the future (see point above) [line 730-738 of the manuscript with tracked changes].

Figure 1 - the heading should stand alone from the text so again we should know where these data were collected too.

  • We are confused by this comment. The caption for Figure 1 reads “Figure 1. Dataset available and sample sizes for different subsets of the data; cases reported to Wildlife Victoria between 2010 and 2019.”, which states where the data were collected. We checked and all other captions for figures and tables do mention when and where the data were collected. Could you please clarify what we have missed?

Line 120 - I guess this should be 6.7 million?

  • Good point. Fixed now [line 148 of the manuscript with tracked changes].

Paragraph 202 - in my experience this journal has common names of species lower case.

  • Fixed throughout. Thank you for pointing this out [line 265-272, and throughout] of the manuscript with tracked changes].

Line 249 - For me a chi square test is not appropriate for such a huge dataset. It would be much more interesting to use generalised linear mixed models where you could combine temporal data, like year; area of the conflict; taxonomic group etc. Here we could see interactions between the variables rather than a long list of descriptive stats. It is also not typical to report the observed and expected values. For degrees of freedom of one, you also need to do post hoc tests.

  • We agree that this kind of rich datasets can be analysed in a much more complex manner and hope to be able to perform to kind of analyses to answer questions about threats to wildlife in the state of Victoria. However, instead of providing a complete analysis of the data and found complex patterns and interactions, we rather focus on the potential this kind of data has for managers; this approach is relevant to stakeholders outside of Victoria or Australia. As stated above, we don’t believe the land managers who could benefit from this data the most (e.g. city councils which might lack resources to collect large amount of meaningful data on human and non-human threats to wildlife) would have access to these kinds of statistical tools and we do not want to discourage them for using this kind of data. We have now removed observed and expected values.

However, we are confused about the point mentioning post hoc tests. For example, if we compare two categories to know if the number of cases differ from Greater Melbourne and outside Greater Melbourne, we obtained a degree of freedom of 1 (number of categories – 1) but there are not then multiple categories to compare (like we do with the other tests in which df is more than 1, and then we inspect the residuals to see where the main differences lie). However, in this example, we now have added a test to explore if there were differences in the main cause types between Greater Melbourne and outside Greater Melbourne in case this is what was requested [line 328-329 of the manuscript with tracked changes].

Line 283 - be wary with release to the wild being a positive outcome; this depends if all guidelines were followed, including health checks and post release monitoring for an extended period - release on its own and moving animals from say an urban setting where they know how to live their lives, to a rural setting, where they die almost immediately, would certainly not be positive.

  • We lack detailed data on the outcomes for a lot of the cases in the database – though we know that post release monitoring is not performed often, if at all. This is a very good point. We have now reverted to studying the outcomes for what they are, instead of grouping them in positive and negative categories [line 368-387 of the manuscript with tracked changes]..

The discussion is largely a repetition of the results and you have much more opportunity to discuss the work in the context of the literature - for example line 418 - what are these studies? How were the results consistent? Why?

  • We now address these comments by providing more contex on these studies and explaining specifically how and why the results are consistent [line 525-530 of the manuscript with tracked changes].

Line 429 - what studies can you recommend to help them with this? Has it ever worked?

  • We now include more studies to support our recommendations (e.g. Scheelings and Frifth, 2015; Taylor-Brown et al., 2019) and we include a case study showing success in reducing the negative impacts of human-wildlife interactions on koalas (Seydel et al., 2023) [line 551-587 of the manuscript with tracked changes]. But as per our point that WERS data is underutilised to help mitigate threats to wildlife, there are a lot more studies suggesting what might work, rather than what has worked.

Line 443 - again have other urban residents in other places benefited? What evidence is there for this statement?

  • We have deleted this part in an attempt to shorten the discussion.

Line 459 - again here you reference some studies but do not discuss your results in line with them -instead just reiterating your results.

  • We have rearranged the information and deleted this paragraph. Though we have moved this information sooner and discuss what the studies were and the implication [line 590-594 of the manuscript with tracked changes].

Overall the discussion can be shortened in that the results do not need to be repeated but throughout need to be discussed in the context of the literature to help the practitioners see what changes are possible

  • This is a good point, thank you. We have now deleted some of the parts that were just repeating results and we now discuss results and suggestions on WERS data can assist practioners to mitigating some human-wildlife conflicts. We hope this significantly improves the discussion and the significance of the results to inform management (though we acknowledge the discussion is not shorter despite deleting some results from the discussion, because of the request to add more information from all reviewers).  

Round 2

Reviewer 2 Report

Dear authors, 

Thank you for considering and acknowledging my comments and suggestions. 

Author Response

You are welcome. Thank you for your time!